# Learning to Approximate Uniform Facility Location via Graph Neural Networks

**Chendi Qian** [1]   **Christopher Morris** [1]   **Stefanie Jegelka** [2,3]   **Christian Sohler** [4]

## Abstract

Neural networks, particularly message-passing neural networks (MPNNs), are increasingly used as heuristics for hard combinatorial optimization problems. Yet many learning-based methods rely on supervision, reinforcement learning, or gradient estimators, causing high computational cost, unstable training, or limited guarantees. Classical approximation algorithms provide worst-case guarantees but are non-differentiable and cannot adapt to structure in natural input distributions. We study this tradeoff through Uniform Facility Location (UniFL), a problem with applications in clustering, summarization, logistics, and supply chains. We propose a fully differentiable MPNN that incorporates approximation-algorithmic principles without solver supervision or discrete relaxations. The model has provable approximation guarantees and empirically improves on standard approximation algorithms, narrowing the gap to integer linear programming.

## 1. Introduction

Recent years have seen remarkable progress in applying neural networks to a variety of hard *combinatorial optimization* (CO) problems. Message-passing graph neural networks (MPNNs) (Gilmer et al., 2017; Scarselli et al., 2009) have been particularly popular due to their permutation invariance, scalability, and ability to exploit graph sparsity, making them natural candidates for leveraging structural information across CO instances. MPNNs have been used both as end-to-end heuristic solvers, e.g., Bello et al. (2016); Karalias & Loukas (2020); Khalil et al. (2017), and as learned components inside classical exact solvers, such as branch-and-cut pipelines for integer-linear optimization or SAT solving. In the latter setting, an MPNN replaces node-, variable-, or cut-selection heuristics (Gasse et al., 2019; Khalil et al., 2022; Scavuzzo et al., 2024; Tönshoff & Grohe, 2025); see Cappart et al. (2023) for a survey.

The motivation for combining learning and algorithms is to discover better heuristics and adapt to instance distributions, yielding faster, higher-quality solutions in practice by exploiting recurring structure in the data. However, two challenges repeatedly arise: (i) how to train these models effectively and efficiently, and (ii) how to ensure reliable solution quality.

Regarding training, two core difficulties are common. First, supervision with optimal solutions is expensive because computing such solutions is itself hard. Second, CO objectives are typically discrete and therefore non-differentiable. To handle non-differentiability, several works adopt surrogate gradient estimators (Petersen et al., 2024), including straight-through estimators (Bengio et al., 2013), continuous relaxations based on the Gumbel–softmax trick or concrete distribution (Jang et al., 2017; Maddison et al., 2017), and more advanced gradient-estimation techniques such as I-MLE (Niepert et al., 2021) or SIMPLE (Ahmed et al., 2023). In practice, however, these methods can be brittle or cumbersome to tune. Another line of work (Karalias & Loukas, 2020; Karalias et al., 2022; 2025) trains directly on a continuous extension of the discrete objective, circumventing both supervision and differentiability barriers.

Yet, these approaches, and end-to-end learned solvers more broadly, typically do not provide guarantees on the quality of the returned solutions. This stands in sharp contrast to *approximation algorithms*, which provide polynomial-time algorithms with rigorous worst-case guarantees on solution quality (Vazirani, 2001). Such guarantees imply strong out-of-distribution robustness, since they hold for all inputs. The downside is that classical approximation algorithms are usually distribution-agnostic, i.e., they are designed for worst-case instances and therefore do not directly exploit structural regularities present in realistic data distributions, where substantially better performance may be possible.

Recently, *algorithms with predictions* (Mitzenmacher & Vassilvitskii, 2020) have aimed to incorporate learning-based predictions into discrete algorithms. However, these methods largely treat predictions as black boxes and are typically non-differentiable, making them difficult to integrate into

[1]RWTH Aachen University, Germany [2]Technical University of Munich, Germany [3]Massachusetts Institute of Technology, USA [4]University of Cologne, Germany. Correspondence to: Chendi Qian <chendi.qian@log.rwth-aachen.de>.

*Proceedings of the 43rd International Conference on Machine Learning*, Seoul, South Korea. PMLR 306, 2026. Copyright 2026 by the author(s).

fully differentiable neural pipelines. Another line of work integrates GNNs into classical exact solvers, e.g., (Gasse et al., 2019), thereby providing certificates for the final solutions. However, training such systems can be extremely resource-intensive, since it often requires repeatedly running a full solver—hundreds or thousands of times—to generate supervision signals or to evaluate learned policies. This dependence on repeated calls to expensive optimization routines limits scalability to large graphs and large training sets.

This gap between robust but conservative approximation algorithms and expressive but potentially unstable (or costly) learned solvers motivates us to combine the strengths of both paradigms. Ideally, one seeks architectures that are (i) fully differentiable and easy to train, (ii) unsupervised or weakly supervised to avoid costly solver calls, (iii) equipped with provable approximation guarantees, and (iv) able to exploit distribution-specific structure to improve empirical performance beyond the training set. Achieving these goals simultaneously remains a major open challenge, with very little existing work (Yau et al., 2024).

**Contributions** In this work, we take a step toward this vision for the exemplary, fundamental case of the *uniform facility location problem* (UniFL). The UniFL asks to open a subset of facilities in a given finite metric space, each with uniform opening cost, to serve a set of clients, minimizing the sum of opening and connection costs. It has wide-ranging applications in logistics, supply-chain design, clustering, and network design (Lammersen, 2010). In machine learning, facility location-like objectives have also been used for data summarization (Wei et al., 2014; Bairi et al., 2015). Even in this simplified case, with uniform opening costs, facility location is hard to approximate with a polynomial-time approximation algorithm (Bădoiu et al., 2005). Hence, we propose an MPNN-based framework that integrates approximation-algorithmic insights with differentiable learning, exploiting distributional information, while still providing theoretical performance guarantees; see Figure 1 for a schematic overview of our framework. Concretely,

1. we introduce a novel fully differentiable, unsupervised MPNN architecture tailored to UniFL—designed to mirror key structural components of classical approximation algorithms. In addition, a slight change in the cost function enables end-to-end $k$-means clustering.

2. We show how to initialize the model with parameters that yield *constant-factor approximation guarantees* and can be improved during training—bridging the gap between approximation algorithms and neural network-based heuristics.

3. We show that we can train the architecture on a finite

training set and that it provably generalizes to UniFL instances of the same size.

4. Empirically, we show that architecture outperforms classical, non-learned approximation algorithms and is competitive to state-of-the-art branch-and-cut solvers while generalizing reliably to problem instances significantly larger than those seen during training.

*In summary, our approach embeds approximation-algorithmic structure into a fully differentiable, unsupervised MPNN, yielding a provably approximate method that generalizes well and outperforms classical baselines.*

**Conflict of Interest Disclosure** We declare that we have no specific financial conflicts of interest to disclose related to this work.

## 1.1. Related work

In the following, we discuss related work.

**MPNNs** MPNNs (Gilmer et al., 2017; Scarselli et al., 2009) emerged as the most prominent graph machine learning architecture. Notable instances of this architecture include, e.g., Duvenaud et al. (2015); Hamilton et al. (2017); Kipf & Welling (2017), and Veličković et al. (2018), which can be subsumed under the message-passing framework introduced in Gilmer et al. (2017). In parallel, approaches based on spectral information were introduced in, e.g., Bruna et al. (2014); Defferrard et al. (2016); Gama et al. (2019); Kipf & Welling (2017); Levie et al. (2019), and Monti et al. (2017)—all of which descend from early work in Baskin et al. (1997); Goller & Küchler (1996); Kireev (1995); Merkwirth & Lengauer (2005); Micheli & Sestito (2005); Micheli (2009); Scarselli et al. (2009), and Sperduti & Starita (1997).

**Facility location problem** The metric uncapacitated facility location problem is a classical topic in approximation algorithms (Shmoys et al., 1997; Shmoys, 2000). A long line of work developed constant-factor approximation algorithms (see (Shmoys et al., 1997; Charikar & Guha, 1999; Mahdian et al., 2006; Arya et al., 2004; Gupta & Tangwongsan, 2008; Mettu & Plaxton, 2003; Thorup, 2004; Jain & Vazirani, 2001) for a few representative results) using linear programming-based rounding, primal-dual methods, and local search; see Shmoys (2000) for an overview. The problem is APX-hard, and no polynomial-time algorithm can achieve a factor better than $1.463$ unless $P = NP$ (Guha & Khuller, 1999). In constant-dimensional Euclidean space, there is a polynomial-time approximation scheme that was later extended to metric spaces with bounded doubling dimension. Also, distributed approximation algorithms are known (Moscibroda & Wattenhofer, 2005), including an $\mathcal{O}(1)$-approximation algorithm in the CONGEST model on a clique (Berns et al., 2012). Even the UniFL, where all opening costs are identical, remains NP-hard, yet its simplified

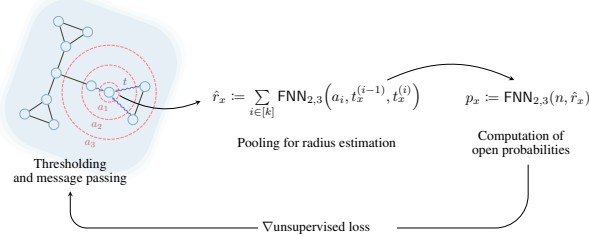

*Figure 1.* Overview of how MPNN probably recovers a near-optimal solution for the UniFL problem. Local message-passing around a given point $x$ is used to compute an estimate of the radius $r_x$, followed by the computation of the opening probabilities, and MPNN's parameters are computed based on an unsupervised loss via the expected cost of the solution.

structure admits efficient local and distributed algorithms. In particular, the radius-based characterization introduced by Mettu & Plaxton (2003), refined in Bădoiu et al. (2005), and the distributed algorithm $\mathcal{O}(1)$-UFL (Gehweiler et al., 2014) yield constant-factor approximation guarantees using only local neighborhood information. Our work builds on these ideas by embedding such local decision rules into a differentiable MPNN architecture, enabling provable approximation guarantees while allowing data-driven refinement of classical algorithmic techniques.

See Appendix B for an extended discussion on related work.

## 2. Background

In the following sections, we introduce the notation and provide the necessary background.

**Basic notations** Let $\mathbb{N} := \{1, 2, \ldots\}$. The set $\mathbb{R}^+$ denotes the set of non-negative real numbers. For $n \in \mathbb{N}$, let $[n] := \{1, \ldots, n\} \subset \mathbb{N}$. We use $\{\!\{\ldots\}\!\}$ to denote multisets, i.e., the generalization of sets allowing for multiple, finitely many instances of each of its elements.

**Graphs** An *(undirected) graph* $G$ is a pair $(V(G), E(G))$ with *finite* sets of *vertices* $V(G)$ and *edges* $E(G) \subseteq \{\{u, v\} \subseteq V(G) \mid u \neq v\}$. The *order* of a graph $G$ is its number $|V(G)|$ of vertices. If not stated otherwise, we set $n := |V(G)|$ and call $G$ an *n-order graph*. Let $\mathcal{G}$ be a set of graphs, then $\mathcal{V}(\mathcal{G})$ denotes the set of pairs $(G, v)$ where $G \in \mathcal{G}$ and $v \in V(G)$. The *neighborhood* of a vertex $v \in V(G)$ is denoted by $N_G(v) := \{u \in V(G) \mid \{v, u\} \in E(G)\}$, where we usually omit the subscript for ease of notation, and the *degree* of a vertex $v$ is $|N_G(v)|$. An *attributed graph* is a pair $(G, a_G)$ with a graph $G$ and an (vertex-)attribute function $a_G : V(G) \to \mathbb{R}^d$, for $d > 0$. The *attribute* or *feature* of $v \in V(G)$ is $a_G(v)$. An *edge-weighted (attributed) graph* is a pair $(G, w_G)$ with a (attributed) graph $G$ and an (edge-)weight function $w_G : E(G) \to \mathbb{R}^+$. For an edge $e \in E(G)$, $w_G(e)$ is the *(edge) weight* of $e$.

Here, we define metric spaces, which play an essential role in the following.

**Metric spaces** Let $\mathcal{X}$ be a set equipped with a *pseudo-metric* $d : \mathcal{X} \times \mathcal{X} \to \mathbb{R}^+$, i.e., $d$ is a function satisfying $d(x, x) = 0$ and $d(x, y) = d(y, x)$ for $x, y \in \mathcal{X}$, and $d(x, y) \leq d(x, z) + d(z, y)$, for $x, y, z \in \mathcal{X}$. The latter property is called the triangle inequality. The pair $(\mathcal{X}, d)$ is called a *pseudo-metric space*. For $(\mathcal{X}, d)$ to be a *metric space*, $d$ additionally needs to satisfy $d(x, y) = 0 \Rightarrow x = y$, for $x, y \in \mathcal{X}$. For $x \in \mathcal{X}$, and $Y \subseteq \mathcal{X}$ let $B_Y(x, r) := \{y \in Y : d(x, y) \leq r\}$. If no ambiguity can arise, we also write $B(x, r)$ instead of $B_{\mathcal{X}}(x, r)$.

See Appendix A for additional details on continuity assumptions, MPNNs, and feed-forward neural networks.

### 2.1. Uniform facility location

Here, we formally introduce the *uniform facility location problem* (UniFL). To that, let $(\mathcal{X}, d)$ be a *finite* metric space, with $|\mathcal{X}| = n$. In the UniFL, the objective is to find a set $F \subseteq \mathcal{X}$ such that

$$\sum_{x \in \mathcal{X}} d(x, F) + |F| \tag{1}$$

is minimized, where $d(x, F) := \min_{y \in F} d(x, y)$. That is, our objective is to find a set $F$ of *facilities* to open, where facilities can be opened at any location $x \in \mathcal{X}$, and, w.l.o.g., each facility costs a uniform 1. The points $x \in \mathcal{X}$ at which we did not open a facility are the *clients*, and they are served by the *closest facility* $f_x$ at cost $d(x, f_x)$. For a given metric space $S$, we denote by $\text{OPT}_S$ the smallest possible value of Equation (1). We remark that we can replace the unit opening cost by an arbitrary value $\lambda > 0$ and minimize $\sum_{x \in \mathcal{X}} d(x, F) + \lambda \cdot |F|$ as we can simply rescale all distances by a factor $1/\lambda$ and obtain the version with opening costs 1. With this formulation, we may interpret the problem as a clustering problem with a regularization term to control the number of clusters.

In the following, we outline some known results that are useful for the UniFL. To that, let $(\mathcal{X}, d)$ be a finite metric space and $x \in \mathcal{X}$. Following Mettu & Plaxton (2003), the *radius* of $x$ is the value $r_x > 0$ such that

$$\sum_{y \in \mathcal{X} \cap B(x, r_x)} r_x - d(y, x) = 1. \tag{2}$$

See Figure 2 in the appendix for an illustration of the radius. Interestingly, knowing the radii of all points in a given metric space $S$ is useful for determining the minimum value $\text{OPT}_S$, as stated in the following result.

**Lemma 1** ((Bădoiu et al., 2005))**.** *Let $S := (\mathcal{X}, d)$ be a finite metric space, then $\sum_{x \in \mathcal{X} \cap B(x, r_x)} r_x \in \Theta(\text{OPT}_S)$.*

Hence, knowing the radius leads to a constant-factor approximation of the smallest possible value of $\text{OPT}_S$.

**Graph representation of UniFL instances** We can straightforwardly encode a given UniFL instance $S := (\mathcal{X}, d)$ as an edge-weighted graph $G_S$ where $V(G_S) := \mathcal{X}$ and $E(G_S) := \{(u, v) : d(u, v) \leq 1\}$ contain edges for every pair of vertices $u$ and $v$ that are at a distance smaller than 1. Note that we remove edges of weight greater than 1 because, following Equation (1), it is cheaper to open a facility than to serve it from a facility at a distance of more than 1. The edge weights $w : E(G_S) \to \mathbb{R}$ of $G_S$ are the distances between the vertices, i.e., $w((u, v)) := d(u, v)$, for $(u, v) \in E(G_S)$. Let $\mathcal{G}_n$ denote the set of graphs encoding UniFL instances of finite metric spaces with $|\mathcal{X}| = n$. If the edge-weights are from a compact subset of $\mathbb{R}$ we write $\mathcal{G}_n^C$.

# 3. An expressive and generalizing MPNN architecture for approximating the UniFL

Here, we devise a simple, scalable MPNN architecture that can represent an $\mathcal{O}(\log(n))$-approximation algorithm for the UniFL with input size $n$ while being trainable in an unsupervised setting. Subsequently, we extend the algorithm to a constant-factor approximation algorithm for the UniFL. We start by devising a distributed approximation algorithm for UniFL and neuralize it, while retaining the ability to recover the approximation factor for the non-learned algorithm. Finally, we show that a finite training dataset is sufficient to learn parameters that simulate the algorithm on the UniFL instance of arbitrary input size.

## 3.1. A simple distributed algorithm for the UniFL

In the following, we derive a distributed $\mathcal{O}(\log(n))$-approximation algorithm for the UniFL, which essentially leverages Lemma 1. Subsequently, we extend to a constant-factor approximation algorithm. That is, given a finite metric space $(\mathcal{X}, d)$, we first compute the radius for each $x \in \mathcal{X}$ and then open a facility with probability $\min(1, c \cdot \ln n \cdot r_x)$, for an appropriate constant $c > 0$. Afterward, points $x \in \mathcal{X}$ that do not have a facility opened at a distance strictly smaller than 1 open up a facility themselves; see SimpleUniformFL for the pseudo-code of the algorithm.

SimpleUniformFL$(\mathcal{X}, d)$
1. **for each** $x \in \mathcal{X}$ **do in parallel** compute the radius $r_x$
2. **for each** $x \in \mathcal{X}$ **do in parallel**
    independently open a facility with probability
    $\min(1, c \cdot \ln n \cdot r_x)$ for an appropriate constant $c > 0$
3. Let $F_1$ be the set of open facilities
4. **for each** $x \in \mathcal{X}$ **do in parallel**
5.     **if** there is an open facility $f$ in $F_1$ with $d(x, f) < 1$
    **then** assign $x$ to $f$
6.     **else** open a facility at $x$

7.     Let $F_2$ be the set of facilities opened in line 6
8.     **return** $F_1 \cup F_2$

The following results establish the $\mathcal{O}(\log(n))$-approximation result of the above algorithm SimpleUniformFL.

**Proposition 2.** *Let $S = (\mathcal{X}, d)$ be a UniFL instance, then the algorithm* SimpleUniformFL *with $c \geq 2$ computes a solution to the UniFL for $S$ with expected cost $\mathcal{O}(\log(n)) \cdot \text{OPT}_S$, where $\text{OPT}_S$ denotes the cost of an optimal solution.*

We remark at this point that the distributed algorithm $\mathcal{O}(1)$-UFL in Gehweiler et al. (2014) achieves a constant approximation in expectation. However, our MPNN implementation computes the opening probabilities, and the algorithm's simplicity allows us to estimate the expected cost of the corresponding solution. This will be crucial for our training procedure. We are not aware of a way to do this for the algorithm from (Gehweiler et al., 2014). In Section 3.3, we argue that we can modify the algorithm in such a way that it only assigns a subset of $\mathcal{X}$ to open facilities and can be recursively applied to the remaining instances in such a way that the resulting (combined over the recursive calls) solution is a constant approximation in expectation.

## 3.2. A simple MPNN architecture for UniFL

In the following, we derive a simple MPNN architecture that is expressive enough to simulate SimpleUniformFL; see Figure 1 for a schematic overview. For each point $x$ in the given metric space, following Equation (6) in the appendix, the MPNN aggregates information from the local neighbors to compute a lower bound for the radius $r_x$, which is subsequently used to compute opening probabilities for each point $x$. Unlike SimpleUniformFL, every step of the architecture is parameterized and fully differentiable, allowing for unsupervised end-to-end training on a given instance distribution.

Formally, let $S := (\mathcal{X}, d)$ denote a finite metric space, $x \in \mathcal{X}$ be a client, and, w.l.o.g., we assume $r_x \leq 1$. To compute the radius of $x$, we discretize the range $(0, 1]$ into "bins" and determine into which bin $r_x$ falls. To that, for $k > 0$, let $(a_0, a_1, \ldots, a_k)$ be a tuple of real numbers, a *discretization*, such that $0 = a_0 < a_1 < a_2 < \cdots < a_k = 1$. We can now estimate the radius via the following local aggregation,

$$
t_x^{(i)} := \min \left\{ 1, \sum_{y \in N(x)} \text{reLU}(a_i - d(x, y)) \right\}
$$
$$
= 1 - \text{reLU}\left( 1 - \sum_{y \in N(x)} \text{reLU}(a_i - d(x, y)) \right),
$$

for $i \in [k]$, which we can further parametrize with a two-

layer FNN using ReLU activation functions as follows,

$$t_x^{(i)} \equiv \mathsf{FNN}_{2,3}\Big( \sum_{y \in N(x)} \mathsf{FNN}_{1,3}(a_i, d(x,y)) \Big), \quad (3)$$

where the existence of the required weight assignments is easily verified. The idea is that $t_x^{(i)}$ is equal to 1 if the radius is at least $a_i$. Hence, we compute an estimate of the radius

$$\begin{aligned}
\hat{r}_x &:= \sum_{i \in [k]} a_i \cdot \Big( t_x^{(i-1)} - t_x^{(i)} \Big) \\
&\equiv \sum_{i \in [k]} \mathsf{FNN}_{2,3}\Big( a_i, t_x^{(i-1)}, t_x^{(i)} \Big),
\end{aligned} \quad (4)$$

via parameterized local aggregation, and estimate the probability of making client $x$ a facility via

$$p_x := \min\{1, c \cdot \log(n) \cdot \hat{r}_x\} \equiv \mathsf{FNN}_{2,3}(n, \hat{r}_x).$$

**Training to solve UniFL** In the following, we devise an unsupervised loss that allows training to adapt the parameters of the above MPNN architectures to UniFL instance distributions. To that, let $F_1$ denote the random variable for the set of facilities opened by the algorithm in line 2, and $F_2$ be the random variable for the set of facilities opened by the algorithm in line 7 in the algorithm SIMPLEUNIFORMFL. We first observe that a facility is opened if it belongs to $F_1$ or $F_2$. Let $A_f$ denote the indicator random variable for the event that $f \in F_1$ and let $B_f$ denote the indicator random variable for the event $f \in F_2$, i.e., there is no facility in $F_1$ that has a distance smaller than 1. The expected opening cost can then be written as

$$\mathbb{E}\left[ \sum_{f \in X} A_f + \sum_{f \in X} B_f \right] = \sum_{f \in X} \mathbb{E}[A_f] + \sum_{f \in X} \mathbb{E}[B_f]$$

by linearity of expectation. We further have $\mathbb{E}[A_f] = \Pr[A_f = 1] = p_f$ and $\mathbb{E}[B_f] = \Pr[B_f = 1] = \prod_{x \in X; d(x,f)<1}(1 - p_x)$. Thus, the expected opening cost is

$$\sum_{f \in X} p_f + \sum_{f \in X} \prod_{x \in X; d(x,f)<1} (1 - p_x).$$

Recall that we only connect clients to $F_1$. To obtain the expected connection cost at a point $x \in X$, we observe that if there is no $f \in F_1$ within distance 1 of $x$, the facility opens and incurs no connection cost. If this is not the case, we connect to the nearest open facility from $F$. A facility $f$ is the nearest open facility to a point $x$, if $f$ is open and all facilities that are closer to $f$ are closed. The latter happens with probability $p_f \cdot \prod_{z \in X; d(x,z)<d(x,f)}(1 - p_z)$. Hence,

overall, we get the following expected costs

$$\begin{aligned}
\mathbb{E}[\mathrm{cost}(X, F)] &= \sum_{f \in X} p_f + \sum_{f \in X} \prod_{x \in X; d(x,f)<1} (1 - p_x) \\
&+ \sum_{x \in X} \sum_{f \in X; d(x,f)<1} \mathrm{dist}(x, f) \cdot p_f \\
&\cdot \prod_{z \in X; d(x,z)<d(x,f)} (1 - p_z),
\end{aligned} \quad (5)$$

where we assume that all pairwise distances are distinct. For a finite metric space $(\mathcal{X}, d)$ with $n = |\mathcal{X}|$ and the corresponding graph representation $G_S$ having at most degree $d$, the running time for evaluating the above loss is in $\mathcal{O}(nd^2)$.

**Approximation guarantees** In the following, we show that there are parameters for the above architecture such that we can recover the $\mathcal{O}(\log(n))$-factor approximation of the Proposition 2 for the non-learned algorithm SIMPLEUNIFORMFL. Precisely, we show the following result.

**Proposition 3.** *For every $n \in \mathbb{N}$, there exists a $k \in \mathbb{N}$, a discretization $0 = a_0 < a_1 < \cdots < a_k = 1$, and an MPNN architecture $\mathsf{mpnn}_{\boldsymbol{\theta}} : \mathcal{V}(\mathcal{G}_n) \to [0,1]$ parameterized by $\boldsymbol{\theta}$, following Section 3.2, and a parameter assignment $\boldsymbol{\theta}^*$ such that for every UniFL instance $S := (\mathcal{X}, d)$ with $|\mathcal{X}| = n$, the MPNN $\mathsf{mpnn}_{\boldsymbol{\theta}^*}$ outputs vertex-wise probabilities*

$$p_v := \mathsf{mpnn}_{\boldsymbol{\theta}^*}(G_S, v) \in [0,1], \quad \text{for } v \in V(G_S),$$

*such that*

$$\mathbb{E}\big[\mathrm{cost}(\mathcal{X}, p)\big] \leq \mathcal{O}(\log(n)) \cdot \mathrm{Opt}_S,$$

*where $\mathrm{cost}(X, p)$ denotes the facility-opening and connection cost obtained by applying the standard SimpleUniformFL post-processing (line 4–6 in SIMPLEUNIFORMFL) using the probabilities $\{p_x\}_{x \in X}$. In particular, the $\mathsf{mpnn}_{\boldsymbol{\theta}^*}$ realizes an $\mathcal{O}(\log(n))$-approximation algorithm for the UniFL on all instances of order $n$.*

For our setting where the algorithm is constrained to compute opening probabilities for the facilities and the solution is obtained by opening the facilities independently with these probabilities, we show that there cannot exist a constant-depth (deterministic) MPNN architecture that achieves an approximation factor of $\mathcal{O}(\log(n))$.

**Proposition 4.** *Let $\mathsf{mpnn}$ be an arbitrary (deterministic) MPNN that*

1. *receives as input an edge-weighted graph $G_S$ that encodes an instance $\mathcal{S} := (\mathcal{X}, d)$ of the uniform facility location problem, and*

2. *computes opening probabilities $p_v$ for every vertex $v \in V(G_S)$.*

*Then there is $n_0$ such that for every $n \geq n_0$ there exists an instance finite metric space $S := (\mathcal{X}, d), with |\mathcal{X}| = n$, such that*

$$\mathbb{E}\left[|F| + \sum_{p \in X} \min\{1, \mathrm{dist}(p, F)\}\right] \geq \frac{\ln(n)}{2} \cdot \mathrm{Opt}_S.$$

### 3.3. Recursively solving UniFL with GNNs

Above, we showed that there exists a constant depth and width MPNN architecture that leads to a $\mathcal{O}(\log(n))$-factor approximation for the UniFL. In the following, we extend this to constant-factor approximation. To that, we first devise a discrete constant-factor approximation that recursively applies SIMPLEUNIFORMFL. To that, we will slightly modify our approach from the previous section, allowing us to leave some facilities unassigned. The sets of open and unassigned facilities are returned. The algorithm may therefore also receive a set of facilities as input, expected to come from an earlier invocation; see UNIFORMFLRECURSIONSTART for the algorithm's pseudo-code.

UNIFORMFLRECURSIONSTART($\mathcal{X}, d$)
1.     $F = \emptyset; X = \mathcal{X}$
2.     **while** $X \neq \emptyset$ **do**
3.          $S, R = $ RECURSIVEUNIFORMFL$(X, F, d)$
4.          $F = F \cup S$
5.          $X = R$
6.     **return** $F$

RECURSIVEUNIFORMFL($\mathcal{X}, F, d$)
1. **for each** $x \in \mathcal{X}$ **do in parallel** compute the radius $r_x$
2. **for each** $x \in \mathcal{X}$ **do in parallel**
3.     independently open a facility at $x$ with probability $\min\{1, c \cdot d(x, F), c \cdot r_x\}$
       for an appr. constant $c > 0$
4. Let $F$ be the set of open facilities
5. $R = \emptyset$
6. **for each** $x \in \mathcal{X}$ **do in parallel**
7.     **if** there is a facility $f$ in $F$ with $d(x, f) \leq 6r_x$
       **then** assign $x$ to the closest $f \in F$
8.     **else** $R = R \cup \{x\}$
9. **return** $F, R$

### 3.4. Learning to solve UniFL from finite data

While Section 3.2 shows that there exist parameters such that an MPNN can achieve an $\mathcal{O}(\log(n))$-factor approximation to the UniFL, it is unclear how to learn a generalizing parameter assignment that works for all metric spaces of a given size. Hence, the following result show that for every size $n \in \mathbb{N}$, there exists an MPNN, a finite training set, and a regularization term such that minimizing the loss and the

regularization term leads to a parameter assignment that recovers the opening probabilities of the MPNN architecture devised in Proposition 3, leading to an $\mathcal{O}(\log(n))$-factor approximation algorithm for any finite metric space of size $n$.

**Proposition 5.** *For every $n \in \mathbb{N}$, there exists an MPNN architecture $\mathsf{mpnn}_{\boldsymbol{\theta}}^{(n)} \colon \mathcal{V}(\mathcal{G}_n^{\mathrm{C}}) \to [0,1]$, with parameters $\boldsymbol{\theta}$, such that for $\varepsilon \in (0,1)$, there is $\varepsilon' > 0$, and a finite training datasets $T_{n,\varepsilon} \subset \mathcal{V}(\mathcal{G}_n^{\mathrm{C}}) \times \mathbb{R}$ consisting of pairs $((G, v), p_v)$, where $G \in \mathcal{G}_n^{\mathrm{C}}$ is a finite, edge-weighted graph encoding an UFL instance, $v \in V(G)$, and $p_v$ is the corresponding opening probability according to Proposition 3, and a differentiable regularization term $r_n$ such that, such that*

$$\frac{1}{|T_{n,\varepsilon}|} \sum_{((G,v),p_v) \in T_n} |\mathsf{mpnn}_{\boldsymbol{\theta}}^{(n)}(G, v) - p_v| + r_n(\boldsymbol{\theta}) \leq \varepsilon'$$

*implies*

$$\|\boldsymbol{p}_{G_S} - h_{\boldsymbol{\theta}}(G_S)\| \leq \varepsilon,$$

*for all $G_S \in \mathcal{G}_n^{\mathrm{C}}$ encoding UniFL instances.*

## 4. Experimental study

Here, we empirically assess the extent to which our theoretical results translate into empirical results and whether we can achieve improved empirical approximation ratios relative to the theoretical worst-case bounds. Concretely, we answer the following questions.

**Q1** Does the MPNN architecture from Section 3.2 lead to improved approximation compared to the non-learned SIMPLEUNIFORMFL?

**Q2** Do the size generalization results of Proposition 5 translate into small generalization error on larger instances than seen during training?

**Q3** Can the MPNN architecture also compete with scalable clustering methods such as KMeans++ and KMedoids++?

**Data generation** We generate synthetic random geometric graphs to evaluate the model's performance with 2, 5, or 10 dimensions. Each graph consists of 1000 vertices. Node positions are sampled from a Gaussian mixture model with 100 isotropic Gaussian components and random centroids. We construct the graph structure by creating an edge $(u, v)$ within a unit Euclidean distance. Additionally, self-loops are added to all vertices to capture situations in which a facility serves itself. We generate 10 000 such graphs for each dataset, splitting them into training, validation, and test sets with a 8:1:1 ratio. Furthermore, to evaluate size generalization, we generate additional test sets with graph sizes ranging from 2000 to 10 000 vertices, with the same Euclidean dimensions and an average node degree approximately equal to that of the training set. Detailed statistics,

including node counts and average degrees, are provided in Table 8.

To assess applicability to realistic structures, we utilize the city map dataset introduced by Liang et al. (2025). This collection consists of four large-scale graphs representing metropolitan road networks, where vertices correspond to junctions and edges represent roads weighted by their lengths. We preprocess these graphs by normalizing the edge weights so that the maximum road length in the network is 1. Notably, these graphs are not strictly geometric, as edge weights may violate the triangle inequality due to the nonlinear geometry of roads (e.g., detours). However, the dataset remains valuable for real-world applicability, where optimal facility location is constrained by realistic edge weights rather than ideal Euclidean distance.

**Experimental protocol** We formulate the UniFL as an integer-linear program (see Appendix D for details), to compute the optimal solution. Besides, we leverage the algorithms SIMPLEUNIFORMFL and UNIFORMFLRECUR-SIONSTART as baselines. They are fully stochastic and serve as a non-learnable counterpart of our MPNN architectures. The constant hyperparameter $c$ of them is tuned via grid search. Furthermore, we include Algorithm 3.1 ($\mathcal{O}(1)$-UFL) in Gehweiler et al. (2014) as a tuning-free baseline. The baselines are all evaluated by averaging 1000 samples with fixed seeds per instance.

For the neural method, we train an MPNN using the proposed unsupervised loss in Equation (5) with 5 different seeds. During inference, the trained model guides sampling in SIMPLEUNIFORMFL, and we also report the expected cost over 1000 samples. To evaluate performance on the $k$-Means objective, we introduce a variant based on Equation (5), replacing the distance in the last term with squared Euclidean distance. We compare this against standard $k$-Means and $k$-Medoids baselines, fixing the number of clusters $k$ to the MPNN's predicted facility count to ensure a fair comparison.

We report the cost components and the optimality ratio relative to the ILP solver. Computational efficiency is measured as the wall-clock time for a single solution generation. That is, for baselines, we compute the radii, while for the MPNN-guided approach, we measure the MPNN inference time, plus one sampling step. Experiments were conducted on an NVIDIA L40S GPU (for neural and stochastic methods) and an Intel® Xeon® Silver 4510 CPU (for ILP and $k$-Means). A detailed description of hyperparameters, architecture, and experiment protocol is provided in Appendix D.

### 4.1. Results and discussion

Here, we present results that answer questions **Q1** to **Q3**.

**Approximations ratios and computation times (Q1)** To

*Table 1.* Performance comparison on synthetic geometric graphs in various dimensions. Results of stochastic methods are averaged over 1000 samples. Besides, MPNN results show the mean $\pm$ standard deviation over five training runs.

| Candidate | Cost | Geo-1000-2 | Geo-1000-5 | Geo-1000-10 | Geo-1000-10-sparse | Geo-1000-10-dense |
|---|---|---|---|---|---|---|
| Solver | Open | 366.302 | 369.884 | 291.488 | 686.578 | 158.483 |
| | Con. | 279.827 | 403.072 | 520.611 | 265.496 | 509.294 |
| | Total | 646.129 | 772.956 | 812.099 | 952.074 | 667.777 |
| | Time | 0.263 | 0.299 | 0.689 | 0.089 | 8.179 |
| SimpleUFL | Open | 512.082 | 523.173 | 436.602 | 816.256 | 243.329 |
| | Con. | 241.435 | 320.425 | 431.908 | 155.263 | 499.281 |
| | Total | 753.517 | 843.598 | 868.511 | 971.519 | 742.610 |
| | Time | 0.001 | 0.001 | 0.001 | 0.001 | 0.001 |
| | Ratio | 1.166 | 1.091 | 1.069 | 1.020 | 1.112 |
| RecursiveUFL | Open | 350.491 | 414.067 | 367.162 | 734.623 | 221.263 |
| | Con. | 367.832 | 415.522 | 496.749 | 227.582 | 520.609 |
| | Total | 718.323 | 829.589 | 863.912 | 962.205 | 741.872 |
| | Time | 0.015 | 0.006 | 0.004 | 0.014 | 0.003 |
| | Ratio | 1.112 | 1.073 | 1.064 | 1.010 | 1.110 |
| $\mathcal{O}(1)$-UFL | Open | 613.904 | 715.915 | 731.670 | 878.267 | 653.293 |
| | Con. | 148.977 | 168.827 | 191.341 | 102.478 | 190.738 |
| | Total | 762.881 | 884.742 | 923.011 | 980.746 | 844.032 |
| | Time | 0.001 | 0.001 | 0.001 | 0.001 | 0.001 |
| | Ratio | 1.181 | 1.144 | 1.137 | 1.030 | 1.263 |
| **MPNN** | Open | 380.816±0.761 | 381.572±0.070 | 298.689±1.334 | 711.482±0.182 | 159.510±2.513 |
| | Con. | 271.317±0.735 | 394.342±0.072 | 515.191±1.301 | 243.491±0.184 | 510.431±2.569 |
| | Total | 652.133±0.025 | 775.914±0.015 | 813.879±0.048 | 954.973±0.003 | 669.942±0.056 |
| | Time | 0.004±0.000 | 0.004±0.000 | 0.004±0.000 | 0.004±0.000 | 0.006±0.000 |
| | Ratio | **1.009**±0.000 | **1.003**±0.000 | **1.002**±0.000 | **1.003**±0.000 | **1.003**±0.000 |

answer **Q1**, the quantitative results on synthetic geometric graphs are presented in Table 1, detailing the facility opening cost, connection cost, total cost, optimality ratio, and execution time. As expected, the stochastic heuristics SIMPLEUNIFORMFL and $\mathcal{O}(1)$-UFL (Gehweiler et al., 2014) exhibit higher costs. A closer look at the cost reveals that $\mathcal{O}(1)$-UFL (Gehweiler et al., 2014) tends to open more facilities than necessary, thereby drastically reducing connection costs but increasing total costs. The recursive refinement in UNIFORMFLRECURSIONSTART mitigates this, achieving ratios mostly within the 10% theoretical bound in Lemma 12. Slight deviations are observed in both the 2D and dense 10D datasets. Most notably, our MPNN architecture consistently outperforms all baselines, achieving near-optimal ratios across all datasets. Crucially, the MPNN also reproduces the balance between opening and connection costs. For instance, on Geo-1000-5, the exact solver gives opening/connection costs at roughly a 370/403 ratio, while our MPNN architectures arrive at a similarly 381/394, indicating the usefulness of the unsupervised loss in Equation (5). Furthermore, the MPNN architecture achieves high precision with negligible computational overhead, with the inference time of 4ms on a GPU.

The performance on the real-world city map graphs is summarized in Table 2. For all four cities, the exact ILP solver failed to converge within 24 hours. In fact, due to the early termination, the solver returned feasible solutions with significantly higher costs than the heuristic methods in several cases, e.g., Paris. Therefore, we cannot show the optimality ratio on these graphs. Among the baselines, RECURSIVEUFL remains the strongest competitor. Our MPNN architecture achieves the lowest total cost across all four datasets. For instance, on the Paris dataset, the GNN reduces the total cost by approximately 41% compared to the solver's incumbent solution.

**Size generalization (Q2)** To answer **Q2**, we test the model

*Table 2.* Performance comparison on real-world graphs. Baseline results are averaged over 1000 samples, while MPNN results show the mean ± standard deviation over five training runs.

| Candidate | Cost | Paris | London | Shanghai | LA |
|---|---|---|---|---|---|
| Solver | Open | 60046 | 247059 | 90967 | 116398 |
| | Con. | 1940.919 | 1649.157 | 548.949 | 1266.352 |
| | Total | 61986.919 | 248708.157 | 91515.949 | 117664.352 |
| | Time | 24h | 24h | 24h | 24h |
| SimpleUFL | Open | 60740.267 | 334712.636 | 104773.305 | 137612.179 |
| | Con. | 508.156 | 983.809 | 386.4197 | 864.918 |
| | Total | 61248.423 | 335696.445 | 105159.725 | 138477.097 |
| | Time | 0.001 | 0.004 | 0.001 | 0.002 |
| RecursiveUFL | Open | 45421.176 | 255379.607 | 79582.307 | 104533.533 |
| | Con. | 779.963 | 1525.648 | 583.267 | 1326.325 |
| | Total | 46201.139 | 256905.256 | 80165.574 | 105859.858 |
| | Time | 0.019 | 0.022 | 0.017 | 0.017 |
| $\mathcal{O}(1)$-UFL | Open | 49408.523 | 271161.191 | 85041.974 | 111659.400 |
| | Con. | 675.359 | 1386.709 | 525.932 | 1191.649 |
| | Total | 50083.882 | 272547.900 | 85567.906 | 112851.049 |
| | Time | 0.067 | 0.379 | 0.113 | 0.146 |
| **MPNN** | Open | 35909.008±280.415 | 185331.256±607.447 | 60728.737±447.568 | 77842.859±317.723 |
| | Con. | 779.735±6.160 | 2160.849±19.504 | 801.318±11.597 | 1757.799±6.267 |
| | Total | **36688.742**±281.216 | **187492.106**±591.426 | **61530.055**±436.448 | **79600.658**±314.253 |
| | Time | 0.078±0.000 | 0.390±0.000 | 0.125±0.001 | 0.166±0.000 |

trained on 1000-vertex graphs on larger instances up to 10 000 vertices without retraining. As shown in Table 3 and Table 10, the approach exhibits remarkable robustness. The optimality ratio remains nearly constant, i.e., on 2D space increasing merely from 1.009 on the training size to 1.012 on graphs 10× larger, and on 10D space 1.002 to 1.003. This indicates that the MPNN successfully learns the data distribution and generalizes effectively to larger-scale instances.

*Table 3.* Results on size generalization, shown in mean ± standard deviations over five runs. MPNN architectures are trained on Geo-1000-10 and tested on larger instances of 10D Euclidean space.

| Candidate | Cost | Geo-1000-10 | Geo-2000-10 | Geo-3000-10 | Geo-5000-10 | Geo-10000-10 |
|---|---|---|---|---|---|---|
| Solver | Open | 291.488 | 638.320 | 1023.900 | 1914.170 | 3961.250 |
| | Con. | 520.611 | 1020.747 | 1500.204 | 2382.913 | 4681.036 |
| | Total | 812.099 | 1659.067 | 2524.104 | 4297.083 | 8642.286 |
| | Time | 0.689 | 2.675 | 5.158 | 12.053 | 55.200 |
| **MPNN** | Open | 298.689±1.334 | 653.247±2.835 | 1050.268±4.668 | 1987.857±16.523 | 4106.202±33.508 |
| | Con. | 515.191±1.301 | 1009.599±2.807 | 1479.893±4.669 | 2322.339±15.082 | 4563.699±31.378 |
| | Total | 813.879±0.048 | 1662.84762±0.135 | 2530.162±0.141 | 4310.197±1.452 | 8669.901±2.278 |
| | Time | 0.004±0.000 | 0.006±0.000 | 0.006±0.000 | 0.007±0.000 | 0.009±0.000 |
| | Ratio | 1.002±0.000 | 1.002±0.000 | 1.002±0.000 | 1.003±0.000 | 1.003±0.000 |

In addition to generalizing to larger sizes, we conducted additional experiments to evaluate how our MPNN generalizes to graphs with varying average node degrees. We took the model pretrained on our Geo-1000-2 dataset and evaluated it without any fine-tuning on three new test sets featuring increasingly dense neighborhoods, Geo-1000-2-d10, Geo-1000-2-d12, and Geo-1000-2-d17, where the postfix denotes the average node degree. The results are as follows in Table 4:

As the results show, generalizing across varying node degrees is more challenging than scaling to larger graph sizes. While the inference time remains fast and stable (around 0.004s), the optimality gap gradually widens from 1.002 on the training data distribution to 1.124 on the densest test set. This behavior intuitively makes sense: increasing the graph size while holding the average degree constant preserves the local neighborhood structure, allowing the MPNN to rely on familiar local patterns. In contrast, increasing the average

*Table 4.* Results on size generalization, shown in mean ± standard deviations over five runs. MPNN architectures are trained on Geo-1000-2 and tested on higher-degree graphs.

| Candidate | Cost | Geo-1000-2 | Geo-1000-2-d10 | Geo-1000-2-d12 | Geo-1000-2-d17 |
|---|---|---|---|---|---|
| Solver | Open | 366.302 | 297.120 | 261.190 | 225.340 |
| | Con. | 279.827 | 282.319 | 273.323 | 255.202 |
| | Total | 646.129 | 579.439 | 534.513 | 480.542 |
| | Time | 0.263 | 0.540 | 0.748 | 1.158 |
| **MPNN** | Open | 380.816±0.761 | 295.560±2.069 | 253.666±4.807 | 225.457±4.101 |
| | Con. | 271.317±0.735 | 290.288±1.995 | 294.991±4.249 | 314.785±11.337 |
| | Total | 652.133±0.025 | 585.848±0.075 | 548.657±0.785 | 540.243±8.703 |
| | Time | 0.004±0.000 | 0.004±0.000 | 0.004±0.000 | 0.005±0.000 |
| | Ratio | 1.009±0.000 | 1.011±0.000 | 1.026±0.001 | 1.124±0.018 |

node degree fundamentally alters the structural topology, shifting the distribution of the messages aggregated at each node during inference. This observation aligns directly with recent theoretical work Levin et al. (2025).

Drawing inspiration from recent graph foundation models Zhao et al. (2024); Beaini et al. (2023), pre-training an MPNN on massive, structurally diverse datasets could yield better generalization across varying topologies without retraining. We conducted further experiments using a structurally diverse training set. We generated 10 000 mixed samples (800–1200 nodes, with average degrees ranging from 5 to 15), matching our original training data budget used in the other experiments. Evaluating this new pretrained model on our size-generalization in Table 10 yields the following results in Table 5:

*Table 5.* Results on size generalization, shown in mean ± standard deviations over five runs. MPNN architectures are trained on a mixed dataset and tested on larger instances of 2D Euclidean space.

| Candidate | Cost | Geo-1000-2 | Geo-2000-2 | Geo-3000-2 | Geo-5000-2 | Geo-10000-2 |
|---|---|---|---|---|---|---|
| Solver | Open | 366.302 | 745.330 | 1201.360 | 1989.970 | 3980.100 |
| | Con. | 279.827 | 548.501 | 813.603 | 1346.195 | 2677.519 |
| | Total | 646.129 | 1293.831 | 2014.963 | 3336.165 | 6657.619 |
| | Time | 0.263 | 0.831 | 1.329 | 3.630 | 13.971 |
| **MPNN** | Open | 385.499±0.351 | 783.525±0.616 | 1269.237±0.564 | 2101.198±1.773 | 4202.280±5.900 |
| | Con. | 267.098±0.280 | 523.752±0.546 | 769.752±0.415 | 1274.294±1.519 | 2535.738±5.384 |
| | Total | 652.598±0.072 | 1307.277±0.069 | 2038.990±0.149 | 3375.493±0.254 | 6738.018±0.516 |
| | Time | 0.004±0.000 | 0.005±0.000 | 0.006±0.000 | 0.006±0.000 | 0.008±0.000 |
| | Ratio | 1.008±0.000 | 1.010±0.000 | 1.011±0.000 | 1.011±0.000 | 1.012±0.000 |

This demonstrates that training on a mixed dataset perfectly preserves the excellent size-generalization capabilities seen in Table 10.

Crucially, we also evaluated this mixed-trained model on Geo-1000-2-d22 (average degree 22) and Geo-1000-2-d27 (degree 27), and the results are in Table 6. As shown, the mixed training strategy is highly effective: the optimality ratio remains exceptionally stable (1.006 to 1.020) even on higher densities, successfully mitigating the structural generalization degradation observed in our previous single-distribution model.

**Comparison with $k$-Means and $k$-Medoids (Q3)** To answer **Q3**, we evaluate the adaptability of our approach by modifying the MPNN loss to minimize the sum of squared distances, effectively simulating the clustering objective. We

*Table 6.* Results on size generalization, shown in mean $\pm$ standard deviations over five runs. MPNN architectures are trained on a mixed dataset and tested on higher-degree graphs.

| Candidate | Cost | Geo-1000-2-d10 | Geo-1000-2-d12 | Geo-1000-2-d17 | Geo-1000-2-d22 | Geo-1000-2-d27 |
|---|---|---|---|---|---|---|
| Solver | Open | 297.120 | 261.190 | 225.340 | 205.900 | 191.320 |
| | Con. | 282.319 | 273.323 | 255.202 | 243.163 | 233.445 |
| | Total | 579.439 | 534.513 | 480.542 | 449.063 | 424.765 |
| | Time | 0.540 | 0.748 | 1.158 | 1.558 | 1.854 |
| MPNN | Open | $305.514_{\pm 1.009}$ | $263.706_{\pm 0.406}$ | $220.253_{\pm 0.763}$ | $194.649_{\pm 0.643}$ | $176.500_{\pm 0.233}$ |
| | Con. | $277.809_{\pm 0.999}$ | $273.959_{\pm 0.373}$ | $264.343_{\pm 0.690}$ | $260.615_{\pm 0.371}$ | $259.011_{\pm 1.102}$ |
| | Total | $583.324_{\pm 0.010}$ | $537.666_{\pm 0.033}$ | $484.596_{\pm 0.072}$ | $455.264_{\pm 0.271}$ | $435.512_{\pm 0.869}$ |
| | Time | $0.004_{\pm 0.000}$ | $0.004_{\pm 0.000}$ | $0.005_{\pm 0.000}$ | $0.005_{\pm 0.001}$ | $0.005_{\pm 0.000}$ |
| | Ratio | $1.006_{\pm 0.000}$ | $1.006_{\pm 0.000}$ | $1.008_{\pm 0.000}$ | $1.013_{\pm 0.001}$ | $1.020_{\pm 0.001}$ |

compare this variant against: (i) Standard $k$-Means++ and $k$-Medoids++ initialization; (ii) Use the MPNN-predicted facility locations as initialization for $k$-Means and $k$-Medoids. The results in Table 7 reveal a consistent hierarchy in connection cost (inertia): $k$-Means++ $<$ GNN $<$ $k$-Medoids++. The advantage of $k$-Means++ is well known: it minimizes inertia by placing centroids in an arbitrary continuous space, thereby enabling solutions that are not valid for the facility location problem. However, in the discrete setting where both methods are constrained to place centroids on existing nodes, our MPNN approach consistently outperforms $k$-Medoids++. Moreover, when initialized with MPNN-predicted locations, both $k$-Means and $k$-Medoids have shown improvements in inertia in most cases. This confirms that our framework effectively adapts to clustering objectives.

*Table 7.* Comparison between MPNN trained with squared connection cost, $k$-Means, and $k$-Medoids. Results are shown as mean $\pm$ standard deviation across five runs trained on each dataset.

| Candidate | Cost | Geo-1000-2 | Geo-1000-5 | Geo-1000-10 |
|---|---|---|---|---|
| GNN + SimpleUFL | Open | $365.167_{\pm 1.180}$ | $359.636_{\pm 1.272}$ | $260.530_{\pm 2.172}$ |
| | Con. | $162.800_{\pm 1.156}$ | $296.984_{\pm 1.938}$ | $444.445_{\pm 3.195}$ |
| | Time | $0.004_{\pm 0.000}$ | $0.004_{\pm 0.000}$ | $0.004_{\pm 0.000}$ |
| $k$-Means++ | Inertia | $130.161_{\pm 0.947}$ | $223.515_{\pm 1.038}$ | $323.422_{\pm 1.886}$ |
| | Time | $0.042_{\pm 0.001}$ | $0.041_{\pm 0.000}$ | $0.055_{\pm 0.000}$ |
| $k$-Means (MPNN init) | Inertia | $132.438_{\pm 1.077}$ | $213.991_{\pm 0.353}$ | $294.124_{\pm 0.186}$ |
| | Time | $0.007_{\pm 0.001}$ | $0.007_{\pm 0.000}$ | $0.013_{\pm 0.001}$ |
| $k$-Medoids++ | Inertia | $170.904_{\pm 1.258}$ | $312.669_{\pm 1.449}$ | $458.796_{\pm 2.592}$ |
| | Time | $0.028_{\pm 0.000}$ | $0.026_{\pm 0.000}$ | $0.017_{\pm 0.000}$ |
| $k$-Medoids (MPNN init) | Inertia | $161.600_{\pm 1.148}$ | $280.005_{\pm 0.321}$ | $398.592_{\pm 2.074}$ |
| | Time | $0.095_{\pm 0.002}$ | $0.095_{\pm 0.001}$ | $0.081_{\pm 0.001}$ |

## 5. Limitations and future directions

While our MPNN frameworks devise the first *unsupervised* neural architecture for the UniFL, several limitations and open problems remain. That is, our approach, which focuses on the UniFL and inherits structural assumptions from the underlying distributed approximation algorithm. While this is essential for preserving worst-case guarantees, it may also restrict flexibility, i.e., the architecture is tailored to UniFL and may require non-trivial redesign to extend to richer variants such as non-uniform opening costs, capacitated facility location, or settings with additional side constraints.

*Looking forward*, there are several promising directions for future research. First, extending the framework beyond UniFL—e.g., to metric facility location with non-uniform costs, capacitated variants, or related problems such as $k$-median or $k$-center—would broaden its applicability and test its generality, ultimately characterizing the set of problems for which MPNN architectures can learn constant-factor approximations. Secondly, developing tighter theoretical analyses that connect learned message passing to instance-dependent approximation behavior, together with understanding under which conditions variants of gradient descent can successfully minimize the expected loss, would help explain when and why training improves solution quality.

## 6. Conclusion

Here, we introduced an unsupervised, light-weight, and fully differentiable MPNN architecture for UniFL. By mirroring a distributed approximation procedure—using learned messages to estimate informative "radii" and a probabilistic facility-opening mechanism—the architecture can be trained end-to-end without relying on optimal labels or reinforcement learning, while still providing worst-case approximation guarantees. Moreover, our framework provides provable size generalization, i.e., parameters learned on smaller instances transfer to arbitrarily larger graphs at test time. Across experiments, the method improves on standard approximation baselines on natural input distributions and narrows the gap to computationally heavier exact approaches, while remaining scalable. *Overall, our results show that leveraging classical algorithmic ideas can yield* differentiable algorithms with guarantees*, opening the door to a broader class of problems.*

## Impact statement

This paper presents work whose goal is to advance the field of machine learning. There are many potential societal consequences of our work, none of which we feel must be specifically highlighted here.

## Acknowledgments

CM and CQ are partially funded by a DFG Emmy Noether grant (468502433) and RWTH Junior Principal Investigator Fellowship under Germany's Excellence Strategy. SJ acknowledges funding from the Alexander von Humboldt Foundation.

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

# A. Extended background

Here, we introduce additional background material.

**Continuity on metric spaces** Let $(\mathcal{X}, d_{\mathcal{X}})$ and $(\mathcal{Y}, d_{\mathcal{Y}})$ be two pseudo-metric spaces. A function $f \colon \mathcal{X} \to \mathcal{Y}$ is called $c_f$-*Lipschitz continuous* if, for $x, x' \in \mathcal{X}$,

$$d_{\mathcal{Y}}(f(x), f(x')) \leq c_f \cdot d_{\mathcal{X}}(x, x').$$

**Message-passing graph neural networks** Message-passing graph neural networks (MPNNs) learn a $d$-dimensional real-valued vector of each vertex in a graph by aggregating information from neighboring vertices. Following Gilmer et al. (2017), let $G$ be an attributed, edge-weighted graph with initial vertex feature $\boldsymbol{h}_v^{(0)} \in \mathbb{R}^{1 \times d_0}$, $d_0 \in \mathbb{N}$, for $v \in V(G)$. An *MPNN architecture* consists of a composition of $L$ neural network layers for some $L > 0$. In each *layer*, $t \in \mathbb{N}$, we compute a *vertex feature*

$$\boldsymbol{h}_v^{(t)} := \mathsf{UPD}^{(t)}\Big(\boldsymbol{h}_v^{(t-1)}, \mathsf{AGG}^{(t)}\Big(\{\!\!\{(\boldsymbol{h}_v^{(t-1)}, \boldsymbol{h}_u^{(t-1)}, w_G(v, u)) \mid u \in N(v)\}\!\!\}\Big)\Big) \in \mathbb{R}^{1 \times d_t}, \tag{6}$$

$d_t \in \mathbb{N}$, for $v \in V(G)$, where $\mathsf{UPD}^{(t)}$ and $\mathsf{AGG}^{(t)}$ may be parameterized functions, e.g., neural networks. In the case of graph-level tasks, e.g., graph classification, one uses a *readout*, where

$$\boldsymbol{h}_G := \mathsf{READOUT}\Big(\{\!\!\{\boldsymbol{h}_v^{(L)} \mid v \in V(G)\}\!\!\}\Big) \in \mathbb{R}^{1 \times d},$$

to compute a single vectorial representation based on learned vertex features after iteration $L$. Again, READOUT may be a parameterized function.

**Feedforward neural networks** An $L$-layer *feed-forward neural network* (FNN), for $L \in \mathbb{N}$, is a parametric function $\mathsf{FNN}_{\boldsymbol{\theta}}^{(L)} \colon \mathbb{R}^{1 \times d_0} \to \mathbb{R}^{1 \times d}$, $d_0, d > 0$, where $\boldsymbol{\theta} := (\boldsymbol{W}^{(1)}, \boldsymbol{b}^{(1)}, \dots, \boldsymbol{W}^{(L)}, \boldsymbol{b}^{(L)}) \in \boldsymbol{\Theta}$, $\boldsymbol{W}^{(1)} \in \mathbb{R}^{d_0 \times d}$, $\boldsymbol{W}^{(i)} \in \mathbb{R}^{d \times d}$, $\boldsymbol{b}^{(i)} \in \mathbb{R}^{1 \times d}$, for $i \in [L]$, and $\boldsymbol{\Theta}$ is an appropriately chosen set, where

$$\boldsymbol{x} \mapsto \sigma\Big(\cdots\sigma\Big(\sigma\Big(\boldsymbol{x}\boldsymbol{W}^{(1)} + \boldsymbol{b}^{(1)}\Big)\boldsymbol{W}^{(2)} + \boldsymbol{b}^{(2)}\Big)\cdots\boldsymbol{W}^{(L)} + \boldsymbol{b}^{(L)}\Big) \in \mathbb{R}^{1 \times d},$$

for $\boldsymbol{x} \in \mathbb{R}^{1 \times d}$. Here, the function $\sigma \colon \mathbb{R} \to \mathbb{R}$ is an *activation function*, applied point-wisely, e.g., a *rectified linear unit* (ReLU), where $\sigma(x) := \max(0, x)$. We define

$$\mathsf{FNN}_{L, d_0, d} := \Big\{\mathsf{FNN}_{\boldsymbol{\theta}}^{(L)} \colon \mathbb{R}^{1 \times d_0} \to \mathbb{R}^{1 \times d} \mid \boldsymbol{\theta} \in \boldsymbol{\Theta}\Big\}.$$

By default, we assume that activation functions are always ReLU unless otherwise stated. Given a function $f \colon \mathbb{R}^{1 \times d_0} \to \mathbb{R}^{1 \times d}$, we write $f \equiv \mathsf{FNN}_{L, d_0, d}$ if there exists an $L \in \mathbb{N}$ and an $f_\theta \in \mathsf{FNN}_{L, d_0, d}$ such that $f(x) = f_{\boldsymbol{\theta}}(x)$, for $x \in \mathbb{R}^{1 \times d_0}$.

# B. Extended related work

Here, we discuss more related work.

**Expressivity and generalization abilities of MPNNs** The expressivity of an MPNN refers to its ability to express or approximate a broad class of functions over graphs. High expressivity means that the network can represent many such functions. In the literature, MPNN expressivity is typically analyzed from two complementary perspectives, i.e., algorithmic alignment with graph isomorphism tests (Morris et al., 2023) and universal approximation theorems (Azizian & Lelarge, 2021; Böker et al., 2023; Geerts & Reutter, 2022). Works in the first line of research investigate whether an MPNN, with suitably chosen parameters, can distinguish the same pairs of non-isomorphic graphs as the 1-dimensional Weisfeiler–Leman algorithm (Weisfeiler & Leman, 1968) or its more powerful generalization (Cai et al., 1992). An MPNN is said to distinguish two non-isomorphic graphs if it computes different vector-valued representations for them. Foundational results by Morris et al. (2019) and Xu et al. (2019) established that the expressive power of any MPNN is upper-bounded by that of 1-dimensional Weisfeiler–Leman algorithm. The second line of research examines which functions over the domain of graphs can be approximated arbitrarily well by MPNNs (Azizian & Lelarge, 2021; Böker et al., 2023; Chen et al., 2019; Geerts & Reutter, 2022; Maehara & NT, 2019). Compared to expressivity, there is comparatively little work

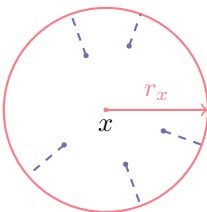

*Figure 2.* Illustration of the radius $r_x$ of the element $x$ according to Equation (2), where the lengths of the dashed lines are summed up.

on the generalization abilities of MPNNs, e.g., see Garg et al. (2020); Scarselli et al. (2018); Vasileiou et al. (2025a;c). In addition, there is a growing body of work on the size-generalization abilities of MPNNs, e.g., (Levin et al., 2025; Maskey et al., 2023; Xu et al., 2021; Le & Jegelka, 2023; Ruiz et al., 2020; Keriven et al., 2020; Levie et al., 2021; Yehudai et al., 2021); see Vasileiou et al. (2025b) for a detailed survey on MPNNs' generalization capabilities.

**(MP)NNs for CO** As mentioned above, there is a growing body of work on using MPNNs for CO, either for directly solving CO problems heuristically, e.g. (Bello et al., 2016; Karalias & Loukas, 2020; Khalil et al., 2017; Selsam et al., 2019) or by integrating them into exact solvers, e.g., in the context of integer-linear optimization (Scavuzzo et al., 2024) or SAT solving (Selsam & Bjørner, 2019; Tönshoff & Grohe, 2025; Wang et al., 2024); see Cappart et al. (2023) for an overview. Recent advances include simulating primal-dual approximation (He & Vitercik, 2025), solving linear optimization problems (Chen et al., 2023; Qian et al., 2024), self-supervised learning for CO problems (Karalias et al., 2025), and provably learning the Bellman–Ford algorithm for shortest paths (Nerem et al., 2025). Recently Yau et al. (2024) showed that MPNN can recover a semidefinite programming-based (optimal) approximation algorithm for maximum constraint-satisfaction problems, such as maximum cut, vertex cover, and maximum 3-SAT. Finally, there is a growing number of works on the capabilities of (graph) transformers (Müller et al., 2024) to execute (graph) algorithms, e.g., de Luca & Fountoulakis (2024); Sanford et al. (2024a;b); Yehudai et al. (2025); Guo et al. (2025). See Cappart et al. (2023) for a survey.

**Approximation algorithms with predictions** A different approach that aims at connecting approximation algorithms and machine learning is *algorithms with predictions* (Mitzenmacher & Vassilvitskii, 2020). In this framework, one seeks to design (online) algorithms that have access to a machine-learning-based prediction oracle that provides advice of unknown quality. If the advice is bad, the goal is to be as good as the best worst-case online algorithm (denoted *robustness*), and if the advice is good, the goal is to be as good as the advice (denote *consistency*). For the online facility location problem (Meyerson, 2001), several prediction-based algorithms are known (Mahdian et al., 2012; Almanza et al., 2021; Jiang et al., 2022; Fotakis et al., 2025). One challenge with algorithms with prediction is often that it is unclear how the predictions will be learned.

## C. Missing proofs

Here, we outline missing proofs from the main paper.

### C.1. Approximation factor of SIMPLEUNIFORMFL

Here, we prove the $\mathcal{O}(\log(n))$-approximation factor of SIMPLEUNIFORMFL.

**Lemma 6.** *Let $S \coloneqq (\mathcal{X}, d)$ be a UniFL instance. Let $x \in \mathcal{X}$ be arbitrary. Then there exists a set $T \subseteq \mathcal{X}$ of size $k$ such that for every $y \in T$ we have $d(y, x) \leq 2r_x$ and $k \geq 1/(2r_y)$.*

*Proof.* Let $x \in \mathcal{X}$ be arbitrary. Define $x^{(0)} = x$ and $x^{(1)} \in B(x^{(0)}, r_{x^{(0)}})$ to be the point with minimum $r_{x^{(1)}}$. If $r_{x^{(1)}} \geq r_{x^{(0)}}/2$ then define $T = B(x, r_{x^{(0)}})$. We know that $k \coloneqq |B(x, r_{x^{(0)}})| \geq 1/r_{x^{(0)}} \geq 1/(2r_{x^{(1)}}) \geq 1/(2r_y)$, for every $y \in T$ which satisfies the second condition of the lemma while the first condition is clearly satisfied.

Otherwise, $r_{x^{(1)}} < r_{x^{(0)}}/2$. In this case, we define $x^{(2)} \in B(x^{(1)}, r_{x^{(1)}})$ to be the point with minimum $r_{x^{(2)}}$ and, more generally, $x^{(i)} \in B(x^{(i-1)}, r_{x^{(i-1)}})$ to be the point with minimum $r_{x^{(i)}}$. As before, we proceed as follows. If $r_{x^{(i)}} \geq r_{x^{(i-1)}}/2$ then the recursion stops and we define $T = B(x^{(i-1)}, r_{x^{(i-1)}})$. Otherwise, we recurse again.

Observe that the recursion eventually stops as the smallest possible radius is $1/n$. Let $x^{(r)}$ be the point that is considered when the recursion stops, and so $T = B(x^{(r-1)}, r_{x^{(r-1)}})$. We know that $k \coloneqq |B(x^{(r-1)}, r_{x^{(r-1)}})| \geq 1/r_{x^{(r-1)}} \geq 1/(2r_{x^{(r)}}) \leq$

$1/(2r_y)$ for every $y \in T$ which implies that every $y \in T$ satisfies the second condition of the lemma. Furthermore, for every $y \in T$ we have $d(x, y) \leq d(x^{(k)}, y) + \sum_{i=1}^{k} d(x^i, x^{i-1}) \leq d(x^{(k)}, y) + r_x \cdot \sum_{i=1}^{k} \frac{1}{2^{j-1}} \leq r_x/2^{k-1} + \sum_{i=1}^{k} \frac{1}{2^{j-1}} = 2r_x$. Hence, the first condition is also satisfied, and the lemma follows. $\qquad\square$

**Lemma 7** (Proposition 2 in the main paper). *Let $S = (\mathcal{X}, d)$ be a UniFL instance, then the algorithm* SIMPLEUNIFORMFL *with $c \geq 2$ computes a solution to the UniFL for $S$ with expected cost $\mathcal{O}(\log(n)) \cdot \mathrm{OPT}_S$, where $\mathrm{OPT}_S$ denotes the cost of an optimal solution.*

*Proof.* For $x \in \mathcal{X}$ let $Y_x$ denotes the indicator random variable for the event that $x \in F_1$. Furthermore, let $Z_x = \min(1, d(x, F))$ denote the random variable for the connection cost, i.e., the distance to the closest open facility (where we assume that this distance is at most 1, since otherwise we open a facility in $F_2$). We first show that by Lemma 1

$$\mathbb{E}\left(\sum_{x \in \mathcal{X}} Y_x\right) = \sum_{x \in \mathcal{X}} \mathbb{E}(Y_x) \leq c \cdot \log n \cdot \sum_{x \in \mathcal{X}} r_x \in \mathcal{O}(\log n) \cdot \mathrm{OPT}_S.$$

In order to estimate $Z_x$ we will first argue that with good probability, there is an open facility within distance $2r_x$, and we pay 1 to open a facility in $x$, if there is no such nearby facility. By Lemma 6 there is a set $T$ of $k$ points such that for every $y \in T$ we have $d(x, y) \leq 2r_x$ and radius $r_y \geq 1/(2k)$. The probability that none of these points has been opened at most $(1 - \min(1, \frac{c \ln n}{2k}))^k$ If $k < \frac{c \ln n}{2}$ this probability is 0, so we assume in the following that $k \geq \frac{c \ln n}{2}$. In this case, we get $(1 - \min(1, \frac{c \ln n}{2k}))^k = (1 - \frac{c \ln n}{2k})^k \leq e^{-k \frac{c \ln n}{2k}} = e^{-\frac{c}{2} \cdot \ln n} = \frac{1}{n^{c/2}}$. Thus,

$$\mathbb{E}\left(\sum_{x \in \mathcal{X}} Z_x\right) = \sum_{x \in \mathcal{X}} \mathbb{E}(Z_x) \leq \sum_{x \in \mathcal{X}} \left(2r_x + \frac{1}{n^{c/2}}\right) \in \mathcal{O}(\mathrm{OPT}_S).$$

The lemma follows from the fact that the expected facility location cost is at most the sum of the expected opening cost and the connection cost. $\qquad\square$

### C.2. Approximation capabilities of the MPNN architecture

Here, we prove the $\mathcal{O}(\log(n))$-approximation factor of the MPNN architecture.

**Proposition 8** (Proposition 3 in the main paper). *For every $n \in \mathbb{N}$, there exists a $k \in \mathbb{N}$, a discretization*

$$0 = a_0 < a_1 < \cdots < a_k = 1,$$

*and an MPNN architecture $\mathsf{mpnn}_{\boldsymbol{\theta}} : \mathcal{V}(\mathcal{G}_n) \to [0,1]$ parameterized by $\boldsymbol{\theta}$, following Section 3.2, and a parameter assignment $\boldsymbol{\theta}^*$ such that for every UniFL instance $S := (\mathcal{X}, d)$ with $|\mathcal{X}| = n$, the MPNN $\mathsf{mpnn}_{\boldsymbol{\theta}^*}$ outputs vertex-wise probabilities*

$$p_v := \mathsf{mpnn}_{\boldsymbol{\theta}^*}(G_S, v) \in [0,1], \quad for \ v \in V(G_S),$$

*such that*

$$\mathbb{E}\big[\mathrm{cost}(\mathcal{X}, p)\big] \leq \mathcal{O}(\log(n)) \cdot \mathrm{Opt}_S,$$

*where $\mathrm{cost}(X, p)$ denotes the facility-opening and connection cost obtained by applying the standard SimpleUniformFL post-processing (line 4–6 in* SIMPLEUNIFORMFL*) using the probabilities $\{p_x\}_{x \in X}$. In particular, the $\mathsf{mpnn}_{\boldsymbol{\theta}^*}$ realizes an $\mathcal{O}(\log(n))$-approximation algorithm for the UniFL on all instances of order $n$.*

*Proof.* We verify the result in three steps: (i) show that the MPNN can approximate the radii, (ii) show that the MPNN can compute the opening probabilities, and (iii) show that we can recover the approximation ratio up to a constant, which we outline below. Let $(\mathcal{X}, d)$ be a finite metric space with $n := |\mathcal{X}|$.

*The MPNN approximates the radii* Recall that the radius $r_x \in (0,1]$, for $x \in \mathcal{X}$, defined as the unique solution of

$$\sum_{y \in X \cap B(x, r_x)} \big(r_x - d(x, y)\big) = 1. \tag{7}$$

By construction, the function

$$\varphi_x(r) := \sum_{y \in X \cap B(x,r)} (r - d(x,y))_+, \qquad (t)_+ := \max\{0, t\},$$

is continuous and strictly increasing in $r \in (0,1]$, and $r_x$ is the unique point with $\varphi_x(r_x) = 1$.

Now, for each bin boundary $a_i$, we define the

$$t_x^{(i)} := \min\left\{1, \sum_{y \in N(x)} (a_i - d(x,y))_+\right\} = 1 - \left(1 - \sum_{y \in N(x)}(a_i - d(x,y))_+\right)_+.$$

By construction, $t_x^{(i)}$ depends only on $x$'s neighbors in $G_S$, and both the inner map $(a_i, d(x,y)) \mapsto (a_i - d(x,y))_+$ and the outer map $z \mapsto \min\{1, z\}$ can be implemented by constant-depth, constant-width FNNs with ReLU activation, which can be verified readily, following the construction in Section 3.2. Hence, an MPNN with sum aggregation and two FNN layers can compute all values $\{t_x^{(i)}\}_{i \in [k]}$ in parallel.

Consider Equation (4), because $\varphi_x(\cdot)$ is strictly increasing and crosses the threshold value 1 exactly once, the sequence $\{t_x^{(i)}\}_{i=0}^k$ changes from values strictly below 1 to values equal to 1 at (or very close to) the true radius $r_x$. With the given discretization, this implies that there exists an index $j \in [k]$ such that

$$a_{j-1} \le r_x \le a_j \quad \text{and} \quad \hat{r}_x \in [a_{j-1}, a_j].$$

Now, let,

$$\Delta := \max_{i \in [k]}(a_i - a_{i-1}).$$

Hence,

$$|\hat{r}_x - r_x| \le \Delta.$$

Consequently, overall,

$$\left|\sum_{x \in X} \hat{r}_x - \sum_{x \in X} r_x\right| \le n\Delta.$$

Hence, by choosing a discretization such that $\Delta$ is sufficiently small, we can make the absolute difference in Appendix C.2 arbitrarily small.

*Computing opening probabilities from radii* The SIMPLEUNIFORMFL algorithm in Section 3.1 opens a facility at $x$ in line 2 with probability

$$p_x^\star := \min\{1, c \log(n)\, r_x\}$$

for a suitable constant $c > 0$. Our MPNN uses the approximate quantities $\hat{r}_x$ and computes

$$p_x := \min\{1, c \log(n)\, \hat{r}_x\} = 1 - (1 - c \log(n)\, \hat{r}_x)_+.$$

Again, the mapping $(\log(n), \hat{r}_x) \mapsto p_x$ is realized by a constant-depth, constant-width FNN with ReLU activation, applied point-wise to the vertex features produced by the above-constructed MPNN, following Section 3.2. Hence, overall, we have constructed the desired MPNN $\mathsf{mpnn}_{\boldsymbol{\theta}^*}$, with parameter assignment $\boldsymbol{\theta}^*$, such that

$$|p_x - p_x^\star| \le c \log(n)|\hat{r}_x - r_x| \le c \log(n)\Delta, \quad \text{for all } x \in \mathcal{X}. \tag{8}$$

Again, by choosing a discretization such that $\Delta$ is sufficiently small, we can arbitrarily closely approximate $p_x^\star$, for $x \in \mathcal{X}$.

*Approximation guarantee* Recall from Section 3.1 that if we denote by $F$ the random set of facilities opened in line 2 of SIMPLEUNIFORMFL, and by $A_f$ and $B_f$ the indicator variables for the events "$f \in F$" and "no facility within distance $< 1$ of $f$ is open after line 2," then the expected opening cost with probabilities $\{p_x\}$ can be written as

$$\mathbb{E}\big[\mathrm{open}(X, p)\big] = \sum_{f \in X} p_f + \sum_{f \in X} \prod_{x \in X := d(x,f)<1} (1 - p_x),$$

and the expected connection cost admits the expression

$$\mathbb{E}\big[\operatorname{conn}(X,p)\big] = \sum_{x \in X} \sum_{f \in X} d(x,f)\, p_f \prod_{z \in X: d(x,z) < d(x,f)} (1 - p_z),$$

so that the total expected cost is

$$\mathbb{E}\big[\operatorname{cost}(X,p)\big] = \mathbb{E}\big[\operatorname{open}(X,p)\big] + \mathbb{E}\big[\operatorname{conn}(X,p)\big].$$

Now let

$$\Phi(p) := \mathbb{E}\big[\operatorname{cost}(X,p)\big].$$

By Equation (8), for every $\varepsilon > 0$, there exists $\Delta_\varepsilon$ such

$$|\Phi(p) - \Phi(p^\star)| \le \varepsilon. \tag{9}$$

On the other hand, Lemma 1 shows that

$$\sum_{x \in X} r_x \in \Theta(\operatorname{Opt}_S),$$

and Proposition 2 applied to SIMPLEUNIFORMFL with the ideal probabilities $\{p_x^\star\}$ yields

$$\mathbb{E}\big[\operatorname{cost}(X,p^\star)\big] \le C \log(n) \cdot \operatorname{Opt}_S,$$

for some $C > 0$. Hence, using Equation (9) and the fact that we can absorb the additive term $\varepsilon$ into the multiplicative constant in front of $\log(n) \cdot \operatorname{Opt}_S$. That is, there exists a universal constant $C_\varepsilon > 0$ such that

$$\mathbb{E}\big[\operatorname{cost}(X,p)\big] \le C_\varepsilon \log(n \cdot \operatorname{Opt}_S.$$

This shows that $M_\theta$ realizes an $O(\log(n))$-approximation algorithm for UniFL on all instances of order $n$, completing the proof. □

**Lemma 9** (Proposition 4 in the main paper). *Let* mpnn *be an arbitrary (deterministic) MPNN that*

1. *receives as input an edge-weighted graph $G_S$ that encodes an instance $S = (X, d)$ of the uniform facility location problem, and*

2. *computes opening probabilities $p_v$ for every vertex $v \in V$.*

*Then there is $n_0$ such that for every $n \ge n_0$ there exists an instance finite metric space $S := (\mathcal{X}, d)$, with $|\mathcal{X}| = n$, such that*

$$\mathbb{E}\left[|F| + \sum_{p \in X} \min\{1, \operatorname{dist}(p, F)\}\right] \ge \frac{\ln(n)}{2} \cdot \operatorname{Opt}_S.$$

*Proof.* We consider an $n$-point metric space $\mathcal{X}$ in which every pair of points from $\mathcal{X}$ has pairwise distance $\varepsilon > 0$ for an arbitrary small $\varepsilon > 0$. An optimal solution (for sufficiently small $\varepsilon > 0$) opens 1 facility and has cost $1 + \varepsilon n$. Since the graph $G_S$ is symmetric and the MPNN mpnn is deterministic, it holds that there is a value $p$ such that $p_v = p$, for $v \in V(G_S)$. Then we get

$$\mathbb{E}\left[|F| + \sum_{p \in X} \min\{1, \operatorname{dist}(p, F)\}\right] \ge pn + (1-p)^n n \ge pn + \left(\frac{1}{e}\right)^{pn} n \ge \frac{\ln n}{2} \cdot \operatorname{Opt}_S.$$

□

### C.3. Analysis of the recursive algorithm

Here, we prove the $\mathcal{O}(1)$-approximation factor of UNIFORMFLRECURSIONSTART.

**Lemma 10.** *Let $(\mathcal{X}, d)$ be a UniFL instance. If we run the algorithm* UNIFORMFLRECURSIONSTART *on $(\mathcal{X}, d)$ then for each recursive call of* RECURSIVEUNIFORMFL *with parameters $\mathcal{X}'$, $F$ and $d$ we have for every $x \in \mathcal{X}'$ that $B_{\mathcal{X}}(x, r_x) \subseteq \mathcal{X}'$ or $d(x, F) \leq 13r_x$, where the radius $r_x$ is defined with respect to set $\mathcal{X}$.*

*Proof.* For $x \in \mathcal{X}$ let $y$ be the first point from $B_{\mathcal{X}}(x, r_x)$ that is assigned to a facility. Consider the corresponding call of RECURSIVEUNIFORM. Let $\mathcal{X}' \subseteq \mathcal{X}$ be the set of points passed to this call. Since $y$ is the first point from $B_{\mathcal{X}}(x, r_x)$ that is assigned to a facility we know that $B_{\mathcal{X}}(x, r_x) \subseteq \mathcal{X}'$. Hence, the value of $r_x$ remains the same for $\mathcal{X}$ and $\mathcal{X}'$. Now consider the current $r_y$ for $y \in B_{\mathcal{X}'}(x, r_x) = B_{\mathcal{X}}(x, r_x)$ that are defined wrt. $\mathcal{X}'$. We claim that $r_y \leq 2r_x$ as otherwise

$$1 = \sum_{p \in B_{\mathcal{X}'}(x, r_x)} r_y - d(p, y) > \sum_{p \in B_{\mathcal{X}'}(x, r_x)} 2r_x - (d(p, x) + d(x, y)) \geq \sum_{p \in B_{\mathcal{X}'}(x, r_x)} r_x - d(p, x) = 1.$$

At the point in time $y$ gets assigned, there is a facility within distance $6r_y$ from $y$ and so there is a facility within distance $13r_x$ from $x$ as $d(x, y) \leq r_x$. During further recursive calls, $x$ will therefore be assigned to a facility within distance at most $13r_x$. $\qquad\square$

**Corollary 11.** *Let $(\mathcal{X}, d)$ be a UniFL instance. If we run the algorithm* UNIFORMFLRECURSIONSTART *on $(\mathcal{X}, d)$ then*

$$\sum_{x \in \mathcal{X}} d(x, F) \in O(\text{OPT}),$$

*where* OPT *denotes the objective value of an optimal solution.*

*Proof.* Follows from Lemma 10 and Lemma 1. $\qquad\square$

We now continue to analyze the opening cost, i.e., $|F|$. We will use Lemma 6

**Lemma 12.** *Consider an arbitrary call to* RECURSIVEUNIFORMFL *with $c \geq 6$ with parameters $(\mathcal{X}, F', d)$. Let $F$ and $R$ denote the returned sets of points. For every $x \in \mathcal{X}$ we have*

$$\Pr(x \in R) \leq 1/10.$$

*Proof.* Let $x \in \mathcal{X}$ be arbitrary. If $d(x, F') \leq 6r_x$ then $x$ will be assigned to an open facility with probability 1. Thus, we can assume that $d(x, F') > 6r_x$. By Lemma 6 there is a set $T$ of $k$ points within distance at most $2r_x$ from $x$ and $k \geq 1/(2r_y)$ for all $y \in T$. We observe that for all $y \in T$ we have $d(y, F) > 4r_x$ by the triangle inequality. By a similar argument (the ball around $y$ cannot contain the ball around $x$) as in the previous proof, each $y \in T$ has $r_y \leq 4r_x$. Hence, $r_y \leq d(y, F)$ and so we select each $y \in T$ with probability at least $\min\{1, c \cdot d(y, F), c \cdot r_y\} = \min\{1, cr_y\}$.

The probability that none of these points has been opened is at most $(1 - \min(1, \frac{c}{2k}))^k$ If $k < \frac{c}{2}$ this probability is 0, so we assume in the following that $k \geq \frac{c}{2}$. In this case, we get $(1 - \min(1, \frac{c}{2k}))^k = (1 - \frac{c}{2k})^k \leq e^{-k\frac{c}{2k}} = e^{-\frac{c}{2}} = \frac{1}{e^3} \leq 1/10$, for $c \geq 6$. $\qquad\square$

Now, the following results show that UNIFORMFLSTART indeed yields a constant-factor approximation for UniFL.

**Lemma 13.** *Consider any call to* UNIFORMFLSTART *with parameters $(\mathcal{X}, F, d)$. Then*

$$\mathbb{E}[|F|] \in \mathcal{O}(\text{OPT}),$$

*where* OPT *denotes the cost of an optimal solution.*

*Proof.* By Lemma 12 we get that the probability that a facility is opened at $x$ in the $i$-th recursive call is at most $(1 - \frac{1}{10})^i \cdot c \cdot d(x, F)$. By Lemma 10 this probability is at most $(1 - \frac{1}{10})^i \cdot 13 \cdot c \cdot r_x$. Let $Y_x^{(i)}$ be the indicator random variable that we

open a facility at $x$ in the $i$-th recursion, and let $Y_x = \sum_{i=1}^{\infty} Y_x^{(i)}$ be the indicator random variable that we open a facility at $x$. Hence, $\mathbb{E}[Y_x] \leq \sum_{i=1}^{\infty} \left(1 - \frac{1}{10}\right)^i \cdot c \cdot d(x, F)$. We get

$$\mathbb{E}\big[|F|\big] = \sum_{x \in \mathcal{X}} \mathbb{E}[Y_x] \in \mathcal{O}\left(\sum_{x \in \mathcal{X}} r_x\right) = \mathcal{O}(\text{OPT}).$$

$\square$

### C.4. Size generalization guarantees

Here, we prove the size generalization guarantees for the MPNN architecture.

**Proposition 14** (Proposition 5 in the main paper). *For every $n \in \mathbb{N}$, there exists an MPNN architecture $\mathsf{mpnn}_{\boldsymbol{\theta}}^{(n)} \colon \mathcal{V}(\mathcal{G}_n^C) \to [0,1]$, with parameters $\boldsymbol{\theta}$, such that for $\varepsilon \in (0,1)$, there is $\varepsilon' > 0$, and a finite training datasets $T_{n,\varepsilon} \subset \mathcal{V}(\mathcal{G}_n^C) \times \mathbb{R}$ consisting of pairs $((G,v), p_v)$, where $G \in \mathcal{G}_n^C$ is a finite, edge-weighted graph encoding an UFL instance, $v \in V(G)$, and $p_v$ is the corresponding opening probability according to Proposition 3, and a differentiable regularization term $r_n$ such that, such that*

$$\frac{1}{|T_{n,\varepsilon}|} \sum_{((G,v), p_v) \in T_n} |\mathsf{mpnn}_{\boldsymbol{\theta}}^{(n)}(G, v) - p_v| + r_n(\boldsymbol{\theta}) \leq \varepsilon'$$

*implies*

$$\|\boldsymbol{p}_{G_S} - h_{\boldsymbol{\theta}}(G_S)\| \leq \varepsilon,$$

*for all $G_S \in \mathcal{G}_n^C$ encoding UniFL instances.*

*Proof sketch.* We essentially use the MPNN architecture from Section 3.2; however, we use order-normalized sum aggregation instead of sum aggregation. That is, given a suitable discretization, we compute

$$t_x^{(i)} := \min\left\{\frac{1}{n}, \frac{1}{n} \sum_{y \in N(x)} \mathsf{reLU}(a_i - d(x,y))\right\} = \frac{1}{n} - \mathsf{reLU}\left(\frac{1}{n} - \sum_{y \in N(x)} \mathsf{reLU}(a_i - d(x,y))\right),$$

for $i \in [k]$. Following Section 3.2, we can further parametrize with a two-layer FNN using ReLU activation functions as follows,

$$t_x^{(i)} \equiv \mathsf{FNN}_{2,3}\left(\frac{1}{n} \sum_{y \in N(x)} \mathsf{FNN}_{1,3}(a_i, d(x,y))\right).$$

The remaining parts of the architecture stay the same. Let $\mathsf{mpnn}_{\boldsymbol{\theta}}^{(n)} \colon \mathcal{V}(\mathcal{G}_n) \to \mathbb{R}$ be the resulting MPNN architecture and $\mathcal{F}_{\boldsymbol{\Theta}} := \{\mathsf{mpnn}_{\boldsymbol{\theta}}^{(n)} \mid \boldsymbol{\theta} \in \boldsymbol{\Theta}\}$, for some appropriately choosen parameter set $\boldsymbol{\Theta}$.

The rest of the proof follows the technique developed in Wittig et al. (2026). That is, implied by (Rauchwerger et al., 2025), assuming the employed FNNs are Lipschitz continuous, there exists a pseudo-metric $d$ on $\mathcal{V}(\mathcal{G}_n)$ such that for each $\mathsf{mpnn}_{\boldsymbol{\theta}}^{(n)} \in \mathcal{F}_{\boldsymbol{\Theta}}$, there is $B_{\theta}$ such that $\mathsf{mpnn}_{\boldsymbol{\theta}}^{(n)}$ is Lipschitz continuous with Lipschitz constant $M_{\boldsymbol{\theta}}$ regarding the pseudo-metric $d$ and the metric space $(\mathbb{R}, |\cdot|)$. In addition, for all $\varepsilon > 0$, there exists a *finite* covering number $K_{\varepsilon} := \mathcal{N}(\mathcal{X}, d, \varepsilon)$.

Observe that, analogous to Proposition 3, we can find parameters $\boldsymbol{\theta}^*$ such that the expected costs are an $\mathcal{O}(\log(n))$-factor approximation. Let $M^*$ be the Lipschitz constant for $\mathsf{mpnn}_{\boldsymbol{\theta}^*}^{(n)}$. Now, set the training set equal to an $r$-cover of the space $(\mathcal{X}, d)$ and set

$$r_n(\boldsymbol{\theta}) := \eta \mathsf{reLU}(M_{\boldsymbol{\theta}} - M^*),$$

for $\eta > 0$. Assume there there is a $\boldsymbol{\theta} \in \boldsymbol{\Theta}$, such for $\mathsf{mpnn}_{\boldsymbol{\theta}}^{(n)}$, the above regularized loss is smaller than $\varepsilon'$. Hence, by exploiting the Lipschitzness and the triangle inequality, we get

$$\|\mathsf{mpnn}_{\boldsymbol{\theta}}^{(n)} - \mathsf{mpnn}_{\boldsymbol{\theta}^*}^{(n)}\|_{\infty} \leq (M_{\boldsymbol{\theta}} + M^*)r + K_r \varepsilon', \tag{10}$$

and that

$$M_{\boldsymbol{\theta}} \leq \frac{1}{\eta}\varepsilon' + M^*. \tag{11}$$

Now, choose $r = \frac{\varepsilon}{6(1+B_{f^*})}$ and $\varepsilon' < \min\left\{\frac{\varepsilon}{3K_r}, \frac{\varepsilon\eta}{6(1+B_{f^*})}\right\}$. By Equation (11), we get

$$M_{\boldsymbol{\theta}} \leq M^* + \frac{\varepsilon'}{\eta} = M^* + \frac{\varepsilon}{6(1+M^*)}.$$

Hence, by using Equation (10), and the assumption $\varepsilon < 1$, and straightforward calculations, we get

$$\|\mathsf{mpnn}_{\boldsymbol{\theta}}^{(n)} - \mathsf{mpnn}_{\boldsymbol{\theta}^*}^{(n)}\|_\infty \leq \varepsilon.$$

$\square$

## D. Details on experiments and additional results

Here, we give additional details on the experiments and discuss additional results.

*Table 8.* Statistics of datasets.

| Name | #node | #degree | #dimension |
|---|---|---|---|
| Geo-1000-2 | 1000 | 6.8 | 2 |
| Geo-2000-2 | 2000 | 7.3 | 2 |
| Geo-3000-2 | 3000 | 6.7 | 2 |
| Geo-5000-2 | 5000 | 6.8 | 2 |
| Geo-10000-2 | 10000 | 7.0 | 2 |
| Geo-1000-5 | 1000 | 7.2 | 5 |
| Geo-1000-10 | 1000 | 10.8 | 10 |
| Geo-2000-10 | 2000 | 10.4 | 10 |
| Geo-3000-10 | 3000 | 10.2 | 10 |
| Geo-5000-10 | 5000 | 9.6 | 10 |
| Geo-10000-10 | 10000 | 10.4 | 10 |
| Geo-1000-10-dense | 1000 | 34.9 | 10 |
| Geo-1000-10-sparse | 1000 | 2.1 | 10 |
| Paris | 114127 | 4.2 | – |
| Shanghai | 183917 | 3.8 | – |
| London | 568795 | 3.7 | – |
| LA | 240587 | 3.8 | – |

**Baselines** We use the optimal solution obtained via an ILP solver as our gold-standard baseline. Specifically, given a UniFL instance $S$, we define binary decision variables $y_i$ for $i \in [n]$ to indicate whether a facility is opened at vertex $i$. Additionally, we define binary assignment variables $e_{ij}$ for $(i,j) \in E(G_S)$ to indicate whether the client at vertex $i$ is assigned to the facility at vertex $j$. The parameter $w_{ij}$ represents the edge distance; notably, the edge set $E(G_S)$ includes self-loops with $w_{ii} = 0$ to allow for self-service. The UniFL problem is formulated as follows,

$$\arg\min_{\boldsymbol{y},\boldsymbol{e}} \quad \sum_{i \in [n]} y_i + \sum_{(i,j) \in E} w_{ij}e_{ij}$$
$$\text{s.t.} \quad e_{ij} \leq y_j$$
$$\sum_{j:\, (i,j) \in E(G_S)} e_{ij} = 1$$
$$y_i, e_{ij} \in \{0,1\}$$

The first constraint ensures that vertex $i$ can be assigned to vertex $j$ only if a facility is open at $j$. The second constraint guarantees that every location is assigned to exactly one facility, potentially itself.

**Experimental protocol** We employ Google's OR-Tools (Perron & Furnon) as the exact ILP solver, imposing a 24-hour time limit per instance. For the $\mathcal{O}(1)$-UFL (Gehweiler et al., 2014) baseline, we repeat the stochastic process with 1000 seeds

and average the result. For the baselines SIMPLEUNIFORMFL and UNIFORMFLRECURSIONSTART, we precompute vertex radii and estimate the expected cost by averaging results over 1000 sampling seeds per instance. To ensure termination in UNIFORMFLRECURSIONSTART, we limit the while-loop to 100 iterations, and if any vertex has no facilities in the neighborhood (including itself) after the loop limit, we force a facility to open at its location. Both baseline methods rely on a single scalar hyperparameter to scale the facility-opening probability. We sweep this parameter using a grid search over 100 candidate values, logarithmically spaced over the interval $[0\,001, 10]$. We report the final performance as the average over the entire test dataset.

For our MPNN approach, we leverage an MPNN with 32 hidden dimensions for all datasets, six layers for geometric graphs, and ten layers for city map datasets. We train it with the unsupervised loss in Equation (5) for at most 1000 epochs and 100 patience for early stopping. The entire training process is repeated five times using independent random seeds. The trained MPNN then guides the sampling algorithm in SIMPLEUNIFORMFL. Specifically, we drop the $c$ constant, predict the facility-opening probability directly using the MPNN, and repeat the sampling 1000 times with different seeds. We aggregate these results by first averaging over the 1000 samples per graph, then averaging over the dataset. Finally, we report the mean and standard deviation of these dataset-level scores across five training seeds. To enable fair comparison across different problem scales, we also report the mean optimality ratio, defined as the average ratio of a method's cost to the exact solution provided by the solver. For detailed hyperparameters, please see to Table 9.

*Table 9.* Hyperparameters of baselines and MPNN approach.

| Name | SimpleUFL | RecurUFL | MPNN layer | MPNN dim. |
|---|---|---|---|---|
| Geo-1000-2 | 0.560 | 0.037 | 6 | 32 |
| Geo-1000-5 | 0.423 | 0.166 | 6 | 32 |
| Geo-1000-10 | 0.351 | 0.242 | 6 | 32 |
| Geo-1000-10-dense | 0.319 | 0.291 | 6 | 32 |
| Geo-1000-10-sparse | 0.423 | 0.031 | 6 | 32 |
| Paris | 1.417 | 0.095 | 10 | 32 |
| Shanghai | 1.417 | 0.095 | 10 | 32 |
| London | 1.292 | 0.086 | 10 | 32 |
| LA | 1.292 | 0.086 | 10 | 32 |

On the geometric graph datasets, we further investigate a variant of our unsupervised approach tailored to the $k$-Means objective. By squaring the distance term $\text{dist}(x, f)$ in the connection-cost component of the loss function Equation (5), we effectively shift the optimization goal to minimizing the sum of squared errors, treating open facilities as cluster centroids. For this comparative study, we train the MPNN using this modified squared-distance loss while keeping all other training protocols identical. During inference, we use the MPNN to guide SIMPLEUNIFORMFL and evaluate the resulting connection cost as the sum of squared distances. To benchmark this approach, we employ standard clustering algorithms $k$-Means and $k$-Medoids (Pedregosa et al., 2011) as baselines. [1] For each trained MPNN, we set the number of clusters $k$ for these baselines equal to the expected number of facilities predicted by the MPNN. We utilize the $k$-Means++ and $k$-Medoids++ initialization, respectively, running 1000 independent initializations per instance and reporting the average result. Furthermore, we use the GNN-predicted facility locations as initializations for $k$-Means and $k$-Medoids. Finally, we report the mean and standard deviation of the performance across the five runs for both the GNN and the clustering baseline; see Algorithm 1 for a reference algorithm.

For a fair comparison of computational efficiency, we report the wall-clock time required to generate a one-shot solution rather than to repeat the sampling. For SIMPLEUNIFORMFL and $\mathcal{O}(1)$-UFL (Gehweiler et al., 2014), we measure the time for radii pre-computation plus one sampling process. Similarly, for our GNN-guided approach, we record the neural network inference time and the time required for a single sampling step. Finally, UNIFORMFLRECURSIONSTART is timed on a single execution, including the necessary while-loops up to a limit of 100.

All stochastic baselines and GNN training and inference are executed on a single NVIDIA L40S GPU. The exact ILP solver, as well as $k$-Means and $k$-Medoids, runs on an Intel® Xeon® Silver 4510 CPU, as it lacks GPU support. We acknowledge that using different hardware platforms makes a direct speed comparison inexact; however, we report raw runtimes for completeness. We note that the ability to leverage GPU acceleration for massive parallelism is a distinct advantage of our MPNN approach.

---

[1]The $k$-Medoids metric is the Euclidean distance; in our case, we adjusted it to the squared Euclidean distance.

**Algorithm 1** Algorithm for clustering comparison. We compare our MPNN prediction against $k$-Means and $k$-Medoids with $k$-Means++ and $k$-Medoids++ initializations, and also with MPNN-predicted facilities as initialization.

---

**Require:** Graph $G = (V, E)$, vertex location matrix $\boldsymbol{X}$, trained MPNN model $\Phi$, repeat count $R$
**Ensure:** Sum squared costs for MPNN, $k$-Means++ and $k$-Medoids++, and MPNN-initialized $k$-Means and $k$-Medoids
 1: Initialize lists $L_{\mathrm{MPNN}}, L_{\mathrm{mean}}, L_{\mathrm{med}}, L_{\mathrm{mean}}^{\mathrm{init}}, L_{\mathrm{med}}^{\mathrm{init}} \leftarrow \emptyset$
 2: $\mathbf{p} \leftarrow \Phi(G)$           ▷ *Predicted vertex opening probability*
 3: **for** $r \leftarrow 1$ **to** $R$ **do**
 4:      **Step 1: MPNN Sampling**
 5:      Sample $\mathbf{y} \sim \mathrm{Bernoulli}(\min(\mathbf{p}, 1))$
 6:      **if** $\exists v$ s.t. $\{N(v) \cup v\} \cap \{u \mid y_u = 1\} = \emptyset$ **then**
 7:         $y_v \leftarrow 1$          ▷ *If $v$ has no facility in neighborhood, force it open*
 8:      $k \leftarrow \sum y_v$
 9:      $\boldsymbol{X}_{\mathrm{init}} \leftarrow \{\mathbf{x}_v \mid y_v = 1\}$          ▷ *MPNN-predicted centroids*
10:      Append MPNN squared distance cost to $L_{\mathrm{MPNN}}$
11:      **Step 2: $k$-Means++ / $k$-Medoids++**
12:      Run $k$-Means with $k$-Means++ initialization; append inertia to $L_{\mathrm{mean}}$
13:      Run $k$-Medoids with $k$-Medoids++ initialization; append inertia to $L_{\mathrm{med}}$
14:      **Step 3: MPNN-guided $k$-Means / $k$-Medoids**
15:      Run $k$-Means with $\boldsymbol{X}_{\mathrm{init}}$ initialization; append inertia to $L_{\mathrm{mean}}^{\mathrm{init}}$
16:      Run $k$-Medoids with $\boldsymbol{X}_{\mathrm{init}}$ initialization; append inertia to $L_{\mathrm{med}}^{\mathrm{init}}$
17: **return** Means of all lists

---

*Table 10.* Results on size generalization, shown in mean $\pm$ standard deviations over five runs. MPNN architectures are trained on Geo-1000-2 and tested on larger instances of 2D Euclidean space.

| Candidate | Cost | Geo-1000-2 | Geo-2000-2 | Geo-3000-2 | Geo-5000-2 | Geo-10000-2 |
|---|---|---|---|---|---|---|
| Solver | Open | 366.302 | 745.330 | 1201.360 | 1989.970 | 3980.100 |
| | Con. | 279.827 | 548.501 | 813.603 | 1346.195 | 2677.519 |
| | Total | 646.129 | 1293.831 | 2014.963 | 3336.165 | 6657.619 |
| | Time | 0.263 | 0.831 | 1.329 | 3.630 | 13.971 |
| **MPNN** | Open | $380.816 \pm_{0.761}$ | $774.047 \pm_{1.065}$ | $1265.818 \pm_{5.796}$ | $2089.730 \pm_{6.018}$ | $4180.888 \pm_{15.306}$ |
| | Con. | $271.317 \pm_{0.735}$ | $532.306 \pm_{1.060}$ | $772.892 \pm_{4.517}$ | $1284.182 \pm_{4.979}$ | $2553.858 \pm_{12.836}$ |
| | Total | $652.133 \pm_{0.025}$ | $1306.354 \pm_{0.012}$ | $2038.710 \pm_{1.282}$ | $3373.912 \pm_{1.065}$ | $6734.747 \pm_{2.488}$ |
| | Time | $0.004 \pm_{0.000}$ | $0.005 \pm_{0.000}$ | $0.006 \pm_{0.000}$ | $0.006 \pm_{0.000}$ | $0.008 \pm_{0.000}$ |
| | Ratio | $1.009 \pm_{0.000}$ | $1.009 \pm_{0.000}$ | $1.011 \pm_{0.000}$ | $1.011 \pm_{0.000}$ | $1.012 \pm_{0.000}$ |

