# OpenReview forum: "Learning to Approximate Uniform Facility Location via Graph Neural Networks"
_ICML.cc/2026/Conference — ICML 2026 regular_

### Official Review · Reviewer_zW31 · 2026-03-12

**Soundness:** 3
**Presentation:** 3
**Significance:** 3
**Originality:** 3
**Overall Recommendation:** 4
**Confidence:** 4

**Summary:**

The authors propose a learning-based heuristics using Message-Passing Neural Networks (MPNNs) for the Uniform Facility Location (UniFL) problem.   Their contributions in the paper include -

1. A novel MPNN architecture that offers a $O(\log n)$-factor approximation guarantee, which can then be refined through data-driven training.
2. Unsupervised Loss Function: A novel loss function based on expected costs allows the model to be trained without expensive optimal solutions as labels.
3. Instance generalization: The research provides a proof that the MPNN can generalize from smaller training instances to arbitrarily large test instances.

**Compliance With Llm Reviewing Policy:**

Affirmed.

**Key Questions For Authors:**

Can the authors please address the generalization of the architecture to other related combinatorial optimization problems?

**Limitations:**

Yes

**Strengths And Weaknesses:**

Strengths:

1. Algorithmic Embedding: By incorporating the radius-based characterization from classical algorithms (e.g., Mettu & Plaxton) into a differentiable pipeline, the authors ensure the model inherits theoretical robustness while gaining the ability to exploit structural regularities in data.
2. Empirical Performance: The MPNN consistently outperforms non-learned approximation baselines (like SIMPLEUNIFORMFL) and narrows the gap to integer linear programming (ILP) solvers while remaining highly scalable.
3. Generalization Robustness: Experiments on graphs up to 10,000 vertices show that the optimality ratio remains nearly constant, demonstrating remarkable size generalization.

Weaknesses:

1. Can the proposed architecture be generalized to other related combinatorial problems like k-Center? The current architecture is specifically tailored to the UniFL problem with uniform opening costs. Extending it to non-uniform costs or capacitated variants will be useful.
2. While the paper proves an $O(\log n)$ approximation factor, it acknowledges that tighter analyses are needed to fully explain why and when gradient descent successfully improves solution quality beyond these worst-case bounds.
3. The size generalization guarantees rely on specific Lipschitz continuity assumptions of the feed-forward networks (FNNs) used within the MPNN.  Maybe this assumption restricts to what problems the architecture can be user for.

---

> ### Author Rebuttal · Authors · 2026-03-28
>
> Thank you for your fair and constructive review.
>
>
> > Can the proposed architecture be generalized to other related combinatorial problems like k-Center? The current architecture is specifically tailored to the UniFL problem with uniform opening costs. Extending it to non-uniform costs or capacitated variants will be useful.
>
> We think that capacitated variants are an interesting future research direction, and we do not have an immediate idea how to extend our approach to them. For other variants, see the further comments below.
>
> > While the paper proves an approximation factor, it acknowledges that tighter analyses are needed to fully explain why and when gradient descent successfully improves solution quality beyond these worst-case bounds.
>
> Yes, we believe that understanding the dynamics of gradient descent is a very promising open research direction.
>
> > The size generalization guarantees rely on specific Lipschitz continuity assumptions of the feed-forward networks (FNNs) used within the MPNN. Maybe this assumption restricts to what problems the architecture can be user for. Can the authors please address the generalization of the architecture to other related combinatorial optimization problems?
>
> It is possible to characterize the aggregation functions for MPNNs that exhibit the finite Lipschitz property. Hence, it is also possible to characterize which other problems can be captured.
>
>
> > Can the authors please address the generalization of the architecture to other related combinatorial optimization problems?
>
> The present architecture can easily be adapted to variants of facility location when the distances are constant powers of a metric (prior work shows that the radius characterization of uniform facility location costs works as well). In the case of k-center clustering or more precisely unit disk cover (as we can influence the number of centers only indirectly by modifying the regularization term), we believe it is possible to obtain a bicriteria polylog n approximation in the following way: Each point counts the number of points inside a disk with twice the unit radius and opens independently with probability proportional to log n / (#points inside the disk). This also leads to a simple loss function that can be efficiently evaluated. For the case of (polynomially bounded) non-uniform opening costs, we believe that one can design a more complex opening procedure that gives a polylog n approximation. The rough idea is to define the radii as in the Mettu-Plaxton paper and then group them into geometrically increasing intervals. Then one applies the uniform opening scheme to each interval.  However, here the loss function will be more complicated as we need to deal with several rounds and will therefore be less suitable for efficient training.
> In summary, our architecture can be adapted to other variants of the problem.
>
> **Please consider raising our score if you are happy with the answer we provided. We are happy to answer any remaining questions.**

---

> > ### Author Rebuttal · Reviewer_zW31 · 2026-04-01
> >
> > Thanks to the authors for acknowledging the interesting directions in which this work can be extended.  The specific nature of the problem tackled in this work and a lack of a clear generalization result in this paper is making me hesitate to raise my score.   I acknowledge it is beyond the scope of a simple rebuttal to address some of the points I raised.

---

> > > ### Author Response · Authors · 2026-04-08
> > >
> > > We agree that the approach is problem-specific. However, it is one of the first architectures to enable unsupervised training, to provide precise theoretical guarantees, and to output an actual solution rather than merely approximate the solution value. Hence, we believe our approach is a stepping stone for connecting GNNs and local or distributed approximation algorithms.

---

### Official Review · Reviewer_xer6 · 2026-03-12

**Soundness:** 3
**Presentation:** 3
**Significance:** 3
**Originality:** 4
**Overall Recommendation:** 4
**Confidence:** 4

**Summary:**

This paper studies how to combine approximation algorithms and graph neural networks for combinatorial optimization. The authors focus on the problem of ''uniform facility location (UniFL)'' and propose a fully differentiable, unsupervised MPNN architecture that mirrors a radius-based distributed approximation procedure: the network estimates local radii, converts them into facility-opening probabilities, and is trained via an explicit expected-cost objective rather than solver supervision or RL. On the theory side, the paper shows that the architecture can realize an $O(\log n)$-approximation, develops a recursive constant-factor variant, and proves a finite-data size-generalization result for fixed input size. On the empirical side, the method substantially improves over non-learned baselines on synthetic geometric instances, generalizes well to larger graphs than those seen during training, and can be adapted to clustering-style objectives.

**Compliance With Llm Reviewing Policy:**

Affirmed.

**Final Justification:**

My questions have been adequately addressed in the rebuttal. However, I still have some concerns, and I remain mildly positive overall on this paper.

**Key Questions For Authors:**

1. Can the authors clarify the relationship between the one-shot MPNN used in the experiments and the recursive constant-factor algorithm in Section 3.3? In particular, was a recursive neural version implemented or evaluated, and if not, what are the main obstacles to doing so?
2. Could the authors clarify the apparent inconsistencies in the dataset statistics table? For example, Table 4 appears to list Geo-10000-2 with `#node = 5000`, and several larger 10D datasets with `#node = 1000`.

**Limitations:**

Yes.

**Strengths And Weaknesses:**

**Pros**

1. The paper proposes a genuinely interesting bridge between approximation algorithms and differentiable GNNs for UniFL.
2. The paper is well-rounded, with detailed theoretical proofs and a comprehensive empirical evaluation.
3. The unsupervised expected-cost objective is elegant and avoids reliance on optimal labels or RL.
4. The paper is overall well-organized and easy to follow.

**Cons**

1. The empirical gains likely stem from the neural network capturing information about the underlying data distribution, but this does not appear to improve the worst-case approximation guarantee.
2. The success of the approach may rely heavily on the locality and problem structure of UniFL, as well as the locality of the underlying algorithm, which may limit its applicability to other graph-based combinatorial optimization problems.

---

> ### Author Rebuttal · Authors · 2026-03-28
>
> Thank you for your fair and constructive review.
>
> > The empirical gains likely stem from the neural network capturing information about the underlying data distribution, but this does not appear to improve the worst-case approximation guarantee.
>
> Yes, in theory, we only show that there exist parameters that can recover the worst-case ratio of the non-learned algorithms. Our experiments show that there are instances and parameter settings for which we can outperform the discrete algorithms, thereby making learning more flexible. An interesting future direction is to show for which classes of instances this holds, thereby improving the average-case approximation for some instance distributions.
>
> > The success of the approach may rely heavily on the locality and problem structure of UniFL, as well as the locality of the underlying algorithm, which may limit its applicability to other graph-based combinatorial optimization problems.
> We agree here; however, many hard problems admit local distributed approximations or sublinear algorithms that can also be expressed via an MPNN.
>
> > Can the authors clarify the relationship between the one-shot MPNN used in the experiments and the recursive constant-factor algorithm in Section 3.3? In particular, was a recursive neural version implemented or evaluated, and if not, what are the main obstacles to doing so?
>
> We thank the reviewer for pointing out this tiny mismatch. In fact, the RecursiveUFL in Section 3.3 is treated as a non-learnable baseline, see e.g., Table 1. On the other hand, the one-shot prediction from the GNN yields near-optimal results. Therefore, the time cost with a recursive GNN refinement, compared with the potential space for improvement, does not pay off. More importantly, we examined the GNN-predicted probabilities; they almost converge to hard 1 and 0; therefore, repeating the sampling process does not make a practical difference.
>
>
> > Could the authors clarify the apparent inconsistencies in the dataset statistics table? For example, Table 4 appears to list Geo-10000-2 with #node = 5000, and several larger 10D datasets with #node = 1000.
>
> We thank the reviewer for careful reading and for catching this error. The inconsistencies in Table 4 arose during manuscript preparation, when node counts from previous rows were inadvertently duplicated. However, please be assured that the reported degree and dimension statistics were accurate and that all experiments were conducted using the correct datasets. The updated table is provided below for your reference.
>
> | Name                | nodes   | degree | dimension |
> |---------------------|---------|--------|-----------|
> | Geo-1000-2	         | 1000	   | 6.8	   | 2         |
> | Geo-2000-2	         | 2000	   | 7.3	   | 2         |
> | Geo-3000-2	         | 3000	   | 6.7	   | 2         |
> | Geo-5000-2	         | 5000	   | 6.8	   | 2         |
> | Geo-10000-2	        | 10000	  | 7.0	   | 2         |
> | Geo-1000-5	         | 1000	   | 7.2	   | 5         |
> | Geo-1000-10	        | 1000	   | 10.8	  | 10        |
> | Geo-2000-10	        | 2000	   | 10.4	  | 10        |
> | Geo-3000-10	        | 3000	   | 10.2	  | 10        |
> | Geo-5000-10	        | 5000	   | 9.6	   | 10        |
> | Geo-10000-10	       | 10000 	 | 10.4	  | 10      |
> | Geo-1000-10-dense   | 1000	   | 34.9	  | 10        |
> | Geo-1000-10-sparse	 | 1000	   | 2.1	   | 10        |
> | Paris	              | 114127	 | 4.2	   | --        |
> | Shanghai	           | 183917	 | 3.8	   | --        |
> | London	             | 568795	 | 3.7	   | --        |
>
>
> **Please consider raising our score if you are happy with the answer we provided. We are happy to answer any remaining questions.**

---

> > ### Author Rebuttal · Reviewer_xer6 · 2026-04-03
> >
> > I appreciate the authors' effort in providing a thorough and thoughtful response, and my questions have been adequately addressed. However, I still have some concerns about the problem-specific nature of the approach and its broader extensibility beyond this setting. Therefore, I maintain my original score.

---

> > > ### Author Response · Authors · 2026-04-08
> > >
> > > We agree that the approach is problem-specific. However, it is one of the first architectures to enable unsupervised training, to provide precise theoretical guarantees, and to output an actual solution rather than merely approximate the solution value. Hence, we believe our approach is a stepping stone for connecting GNNs and local or distributed approximation algorithms.

---

### Official Review · Reviewer_sBxk · 2026-03-12

**Soundness:** 3
**Presentation:** 3
**Significance:** 3
**Originality:** 3
**Overall Recommendation:** 4
**Confidence:** 3

**Summary:**

This paper considers the uniform facility location problem (UniFL), a combinatorial problem known to be NP-Hard but for which many efficient polynomial-time constant-factor approximation algorithms exist. The authors first introduce a simple parallel (per-node) approximation algorithm for the UniFL, before showing that a Message Passing Neural Network (MPNN) architecture can be used to represent it and achieve the same (logarithmic) approximation guarantees. They then propose to recursively solve it to achieve constant approximation guarantees (for the recursive algorithm), and theoretically prove both the approximation guarantees and the MPNN learnability and generalization guarantees (for fixed-size graphs).

Thorough experiments on synthetic and real-world graphs demonstrate the effectiveness and efficiency of the proposed approach, which is empirically able to achieve better approximations at reduced computational cost compared to state-of-the-art heuristics.

**Compliance With Llm Reviewing Policy:**

Affirmed.

**Key Questions For Authors:**

-	Please address the key points raised within my “weaknesses” textbox (regarding proof sketches, notation clarity, unclear claim and reporting MPNN training times).

-	Could you elaborate on the regularization term introduced in Proposition 6? In particular, how should it be chosen in practice, and how was it instantiated in the experiments? A more detailed discussion would help clarify its role and practical impact.

-	Beyond varying the number of vertices, it would be important to understand how the MPNN generalizes to graphs with different structural properties (e.g., average node degree). Since training the MPNN may represent a substantial computational cost, its ability to generalize across broader classes of graph structures is essential for ensuring that the one-time training overhead is effectively outweighed by the fast inference times.

**Limitations:**

yes

**Strengths And Weaknesses:**

The core idea of this paper is to leverage classical algorithmic ideas (through the proposed parallel approximation algorithm) and incorporate them into a neural architecture to enable efficient inference and to exploit the data distribution. The resulting method provides provable constant approximation guarantees and is empirically competitive with the state of the art (often better), making it a good contribution.

Strengths :

-	Overall, the paper is well written. Related works are appropriately discussed, and both theoretical and empirical results are provided to support the main claims

-	The experimental setup is sound, and the results clearly demonstrate that the proposed method (i) offers comparable or better empirical approximation than state-of-the-art approaches and (ii) is very efficient at inference time

-	The ability of the proposed method to run on a GPU is an important advantage, as evidenced by its highly efficient inference times

-	The good generalization of the MPNN to graphs larger than the ones seen during training is also a good indicator of the method’s potential

-	The unsupervised nature of the loss function used to fit the MPNN (i.e., it does not require prior computation of the instance’s optimal solution) is also a significant advantage for real applications

Weaknesses :

-	I would recommend including proof sketches directly after every proposition’s statement. While the full proofs are appropriately provided in the appendices, providing concise sketches in the main text (summarizing the core argument and the pivotal lemmas) will make the paper more self contained and improve readability.

-	I have some minor concerns regarding notation. In Equation (5) (and in the Equation on the bottom left of page 5), why is $X$ used instead of $\mathcal{X}$ (as in the previous paragraphs) to denote the set of locations? Similarly, in Proposition 3, $X$ and $\mathcal{X}$ are used interchangeably. On the same page (in Equation (5) on the text before it), $F$ is not properly introduced (even though I guessed that it contains all facilities opened either in $F_1$ or $F_2$). Same for $h_\theta$ or $\mathbf{p}_{G_S}$ (Proposition 6): while the meaning of these notations can be guessed from the context, they still should be explicitly defined.

-	Some claims seem over‑stated, and I would recommend to either explain why they should be kept as is, or to reformulate them to avoid ambiguities. In particular, the abstract and the conclusion say: “our framework gives provable size generalization abilities”. However, even though generalization abilities are empirically evaluated, there is no theoretical result regarding cross‑size generalization (to the best of my understanding, Proposition 6 only concerns uniform generalization to graphs of the same size).

-	Training times for fitting the MPNN are not reported. While I understand that an MPNN architecture can, in principle, be reused across different graphs (thereby amortizing the training cost), in the experiments presented the MPNNs are trained separately for each dataset. Therefore, the associated training cost should at least be reported and discussed.

Minor :

-	References (Shmoys et al., 1997a) and (Shmoys et al., 1997b) are duplicates

-	Lemma 1: double parenthesis around citation

-	Section 3.4: “Hence, the following result show that” should be “shows”

---

> ### Author Rebuttal · Authors · 2026-03-28
>
> Thank you for your fair and constructive review.
>
> > I would recommend including proof sketches directly after every proposition’s statement. [...]
> Thank you, that is a good suggestion. We will use the extra page in the final paper to give proof sketches in the main paper.
> > I have some minor concerns regarding notation. [...]
>
> Thank you. We will introduce the missing notation and unify the notation.
>
> > Some claims seem over‑stated [...]
>
> We agree; it is true that Proposition 6 holds only for fixed $n$. We will make the discussion and claims more nuanced.
>
> > Training times for fitting the MPNN are not reported.
>
> We thank the reviewers for pointing out the crucial metric. In fact, training MPNN is very efficient on a single NVIDIA L40S GPU, as the message passing is graph sparsity-aware, and the complexity bottleneck of $\O(nd^2)$ lies only in the loss computation. In Table 1, training on the Geo-1000-2 dataset takes around 12.16 seconds per epoch and, in total, up to 3 hours and 22 mins. With the early stopping mechanism, this can be shortened. For the Geo-1000-5 dataset, it is 15.37 seconds per epoch, and for Geo-1000-10, it is 20.89 seconds per epoch.
>
> In Table 3, training is similar: on Geo-1000-2, it is 11.58 seconds per epoch; on Geo-1000-5, it is 14.43 seconds per epoch; and on Geo-1000-10, it is 20.97 seconds per epoch.
>
> > Could you elaborate on the regularization term introduced in Proposition 6? [...]
>
> The regularization term is outlined on line 1027 of the appendix, which essentially enforces that the Lipschitz constants of the MPNN and the target function are close. We will add it to the main paper in the final paper.
>
> > Beyond varying the number of vertices, it would be important to understand how the MPNN generalizes to graphs with different structural properties (e.g., average node degree). [...]
>
> We thank the reviewer for raising this excellent point regarding structural generalization. To address this, we conducted additional experiments to evaluate how our MPNN generalizes to graphs with varying average node degrees. We took the model pretrained on our base dataset and evaluated it without any fine-tuning on three new test sets featuring increasingly dense neighborhoods, detailed below:
>
> | Name              | nodes   | degree | dimension |
> |-------------------|---------|--------|-----------|
> | Geo-1000-2-train	 | 1000	   | 6.8	   | 2         |
> | Geo-1000-2-test1	 | 1000	   | 9.5	   | 2         |
> | Geo-1000-2-test2	 | 1000	   | 12.3	  | 2         |
> | Geo-1000-2-test3	 | 1000	   | 17.1	   | 2         |
>
> The results are as follows:
>
> | Candidate | Cost  | Geo-1000-10-train | Geo-1000-2-test1 | Geo-1000-2-test2 | Geo-1000-2-test3 |
> |-----------|-------|-----------------|------------------|------------------|------------------|
> | Solver    | Open  | 366.302         | 297.120          | 261.190          | 225.340          |
> | Solver    | Con.  | 279.827        | 282.319          | 273.323          | 255.202          |
> | Solver    | Total | 646.129         | 579.439          | 534.513          | 480.542          |
> | Solver    | Time  | 0.263           | 0.540            | 0.748            | 1.158            |
> | MPNN      | Open  | 380.816±0.761   | 295.560±2.069    | 253.666±4.807    | 225.457±4.101    |
> | MPNN      | Con.  | 271.317±0.735   | 290.288±1.995    | 294.991±4.249    | 314.785±11.337   |
> | MPNN      | Total | 652.133±0.025   | 585.848±0.075    | 548.657±0.785    | 540.243±8.703    |
> | MPNN      | Time  | 0.004±0.000     | 0.004±0.000      | 0.004±0.000      | 0.005±0.000      |
> | MPNN      | Ratio | 1.009±0.000     | 1.011±0.000      | 1.026±0.001      | 1.124±0.018      |
>
> As the results show, generalizing across varying node degrees is more challenging than scaling to larger graph sizes (as discussed in Q3 of the main paper). While the inference time remains fast and stable (around 0.004s), the optimality gap gradually widens from 1.002 on the training data distribution to 1.124 on the densest test set.
> This behavior intuitively makes sense: increasing the graph size while holding the average degree constant preserves the local neighborhood structure, allowing the MPNN to rely on familiar local patterns. In contrast, increasing the average node degree fundamentally alters the structural topology, shifting the distribution of the messages aggregated at each node during inference. This observation aligns directly with recent theoretical work [1]. We will include these new experiments and this discussion in the appendix of the revised manuscript.
>
> [1] Levin, Eitan, et al. "On transferring transferability: Towards a theory for size generalization." arXiv preprint.
>
> **Please consider raising our score if you are happy with the answer we provided. We are happy to answer any remaining questions.**

---

> > ### Author Rebuttal · Reviewer_sBxk · 2026-04-03
> >
> > Thank you for your clarifications.
> >
> > Regarding generalization to other graphs with different structural properties, thank you for these additional results. I agree that they are worth reporting in the revised paper, along with a discussion about how this could be improved (maybe training with graphs with varied properties or even modifying graphs at training time in a “data augmentation” manner could help?).
> >
> > Regarding MPNN training, I think these details should be included in the paper. In particular, Tables 1 and 6 should account for them, as otherwise, it biases the comparison with methods that do not require such expensive training prior to being run to solve the UFL instance. Since the method’s generalization to structurally different graphs remains limited, this suggests that handling a sufficiently different problem would require re-training, therefore this cost is not negligible.
> >
> >  From what I understand, training the MPNN for the experiments on synthetic geometric graphs (those in the first three columns of Table 1) takes between 3 and 6 hours (assuming the number of epochs remains the same for Geo-1000-2, Geo-1000-5 and Geo-1000-10) on a high-end GPU. In the meantime, the exact approach (solver) only requires fractions of a second to solve the instance to optimality. For a fair comparison, the training time should be incorporated in the running time reported for the MPNN-based method, divided by the number of inferences (so it is an amortized preprocessing time). Could you also provide these numbers for the real-world graphs (Table 6) please ?

---

> > > ### Author Response · Authors · 2026-04-05
> > >
> > > We thank the reviewer for the constructive follow-up and the valuable suggestions.
> > > # Regarding Generalization and Mitigation Strategies
> > > We fully agree that the degree-generalization results should be included in the revised paper, alongside a discussion on improving out-of-distribution. Drawing inspiration from recent graph foundation models [1][2], pre-training an MPNN on massive, structurally diverse datasets could yield better generalization across varying topologies without retraining.
> > >
> > > We conducted new experiments using a structurally diverse training set. We generated 10000 mixed samples (800–1200 nodes, average degrees varying from 5 to 15), matching our original training data budget in the paper. Evaluating this new pretrained model on our size-generalization in Table 7 yields the following results:
> > >
> > > | Candidate |Cost | Geo-1000-2    | Geo-2000-2     | Geo-3000-2 | Geo-5000-2     | Geo-10000-2    |
> > > |---|---|---|---|---|---|---|
> > > | Solver | Open | 366.302 | 745.330  | 1201.360 | 1989.970  | 3980.100 |
> > > |  |Con. | 279.827  | 548.501  | 813.603 | 1346.195  | 2677.519 |
> > > |  |Total | 646.129 | 1293.831 | 2014.963 | 3336.165  | 6657.619  |
> > > |  |Timing | 0.263 | 0.831| 1.329   | 3.630 | 13.971  |
> > > | MPNN |Open | 385.499±0.351 | 783.525±0.616  | 1269.237±0.564 | 2101.198±1.773 | 4202.280±5.900 |
> > > |  |Con. | 267.098±0.280 | 523.752±0.546  | 769.752±0.415 | 1274.294±1.519 | 2535.738±5.384 |
> > > |   |Total | 652.598±0.072 | 1307.277±0.069 | 2038.990±0.149 | 3375.493±0.254 | 6738.018±0.516 |
> > > |   |Timing | 0.004±0.000  | 0.005±0.000 | 0.006±0.000 | 0.006±0.000  | 0.008±0.000  |
> > > |  |Ratio | 1.008±0.000  | 1.010±0.000 |1.011±0.000 | 1.011±0.000  | 1.012±0.000  |
> > >
> > > __This demonstrates that training on a mixed dataset perfectly preserves the excellent size-generalization capabilities seen in our Table 7__.
> > >
> > > Crucially, we also evaluated this mixed-trained model on two new datasets with higher average degrees:
> > > | Name | nodes | degree | dimension |
> > > |---|---|---|---|
> > > | Geo-1000-2-test4	 | 1000 | 22.5| 2 |
> > > | Geo-1000-2-test5	 | 1000 | 27.1 | 2 |
> > >
> > > | Candidate | Cost | Geo-1000-2-test1 | Geo-1000-2-test2 | Geo-1000-2-test3 | Geo-1000-2-test4 | Geo-1000-2-test5 |
> > > |---|---|---|---|---|---|---|
> > > | Solver | Open | 297.120  | 261.190| 225.340 | 205.900 |191.320|
> > > | Solver | Con. | 282.319  | 273.323 | 255.202 |243.163|233.445|
> > > | Solver | Total | 579.439 | 534.513 | 480.542 |449.063|424.765|
> > > | Solver | Time  | 0.540 | 0.748 | 1.158 | 1.558 |1.854|
> > > | MPNN | Open  | 305.514±1.009 | 263.706±0.406 | 220.253±0.763 | 194.649±0.643 |176.500±0.233|
> > > | MPNN | Con.  | 277.809±0.999 | 273.959±0.373 | 264.343±0.690 | 260.615±0.371 |259.011±1.102|
> > > | MPNN | Total | 583.324±0.010 | 537.666±0.033 | 484.596±0.072 | 455.264±0.271|435.512±0.869|
> > > | MPNN | Time  | 0.004±0.000  | 0.004±0.000 | 0.005±0.000 | 0.005±0.001|0.005±0.000 |
> > > | MPNN | Ratio  | 1.006±0.000 |1.006±0.000 | 1.008±0.000 |1.013±0.001|1.020±0.001|
> > >
> > > As shown above, __the mixed training strategy is highly effective__: the optimality ratio remains exceptionally stable (1.006 to 1.020) even on higher densities, successfully mitigating the structural generalization degradation observed in our previous single-distribution model.
> > >
> > > # Regarding MPNN Training Time and Amortization
> > > We completely agree that reporting the training cost is essential for transparency. The reviewer is entirely correct that for synthetic graphs, the exact solver's sub-second runtime makes a 3-hour MPNN training impractical for a single instance. For the real-world city map datasets, the training times per epoch are highly efficient and scale proportionally with the graph size:
> > >
> > > | City | Paris | London | Shanghai | LA|
> > > | ---| ---|---|---|---|
> > > |  # nodes | 114127 | 568795 | 183917 | 240587 |
> > > | sec/epoch | 0.27 | 1.20 | 0.32 | 0.56 |
> > >
> > > With these per-epoch time, the entire training process completes in under 20 minutes. Compared to the exact ILP solver, which fails to converge and times out after 24 hours, the neural network training cost is highly competitive. Furthermore, __this highlights a distinct advantage of our unsupervised approach for single, massive real-world applications__. Because our method is __unsupervised__, we can treat the training phase itself as a heuristic optimization process, which is superior to a solver timeout.
> > >
> > > Regarding the suggestion to incorporate an amortized training time, we respectfully argue that dividing the training time by a number of inferences may misrepresent the deployment of neural solvers. Because our MPNN generalizes exceptionally well, __including larger graphs and higher degrees__, a single trained model can theoretically be __applied to an unbounded number of unseen instances__. Therefore, any amortized cost calculation would be arbitrary.
> > >
> > > __If our response and clarified scopes satisfactorily resolve your concerns, we would be incredibly grateful if you consider reflecting this in your final score__.
> > >
> > > [1] https://arxiv.org/abs/2405.20445
> > > [2] https://arxiv.org/abs/2310.04292

---

### Decision · Program_Chairs · 2026-04-30

**Decision:**

Accept (regular)

**Comment:**

This paper bridges the gap between GNNs and classic approximation algorithms by proposing a GNN for the uniform facility location problem that happens to have a guaranteed approximation ratio.

The reviewers agree that the idea is interesting, the paper is in general well written, and that the experiments are sound. A potential weakness is that the approach seems very problem-specific, and it is not clear if anything similar can be done with other combinatorial optimization problems.